Resource

# Predictive data-driven modeling of C-terminal tyrosine function in the EGFR signaling network

Jacqueline S Gerritsen[1,2,3], Joseph S Faraguna[1], Rudy Bonavia[4], Frank B Furnari[4,5,6] , Forest M White[1,2,3]

**The epidermal growth factor receptor (EGFR) has been studied extensively because of its critical role in cellular signaling and association with disease. Previous models have elucidated interactions between EGFR and downstream adaptor proteins or showed phenotypes affected by EGFR. However, the link between specific EGFR phosphorylation sites and phenotypic outcomes is still poorly understood. Here, we employed a suite of isogenic cell lines expressing site-specific mutations at each of the EGFR C-terminal phosphorylation sites to interrogate their role in the signaling network and cell biological response to stimulation. Our results demonstrate the resilience of the EGFR network, which was largely similar even in the context of multiple Y-to-F mutations in the EGFR C-terminal tail, while also revealing nodes in the network that have not previously been linked to EGFR signaling. Our data-driven model highlights the signaling network nodes associated with distinct EGF-driven cell responses, including migration, proliferation, and receptor trafficking. Application of this same approach to less-studied RTKs should provide a plethora of novel associations that should lead to an improved understanding of these signaling networks.**

## Introduction

Epidermal growth factor receptor (EGFR) is a receptor tyrosine kinase (RTK) that is overexpressed, mutated or otherwise dysregulated in a large fraction of human cancers (Mitsudomi & Yatabe, 2010). On ligand binding, EGFR assumes an active conformation, dimerizes, and cross-/ auto-phosphorylation occurs on C-terminal tyrosine residues, thus initiating a cascade of signaling events ultimately resulting in cellular phenotypes such as proliferation and migration, among others (Yarden & Sliwkowski, 2001). EGFR signaling has been well characterized in a variety of cell lines, including immediate-early signaling with high temporal resolution and deep characterization of the network-wide effects of EGFR

activation (Zhang et al, 2005; Dengjel et al, 2007; Reddy et al, 2016). Despite these large scale phosphoproteomics studies, mechanistic insights connecting kinase and phosphatase activity with particular EGFR phosphorylation sites and their associated protein–protein interactions have been challenging.

To address this issue and gain a better understanding of how EGFR activity regulates cell biology, there have been extensive efforts to define the association between EGFR C-terminal tyrosine phosphorylation sites (e.g., Y845, Y974, Y992, Y1045, Y1068, Y1086, Y1101, Y1148, Y1173) and recruitment of adaptor or effector proteins. Many of these studies have been performed using in vitro assays, including a landmark study using protein microarrays to identify and quantify the affinity of Src-homology 2 (SH2) and phosphotyrosine-binding domains for peptides containing phosphorylated tyrosine residues from ErbB family members (Jones et al, 2005) and another using tyrosine phosphorylated peptides from EGFR to identify protein binders in cell lysate (Tong et al, 2014). These studies, along with a plethora of cell-based studies, have yielded critical insight for selected EGFR tyrosine phosphorylation sites. For instance, phosphorylated Tyr 992 has been shown to function as a major binding site for 1-phosphatidylinositol 4,5-bisphosphate phosphodiesterase gamma-1(PLCγ-1) (Rotin et al, 1992; Emlet et al, 1997), whereas phosphorylation of Y1045 serves as the primary EGFR-binding site for Cbl, an E3 ubiquitin ligase that mediates receptor ubiquitination, endocytosis, and degradation or recycling (Levkowitz et al, 1999). Phosphorylated residues Y1068 and Y1086 bind the growth factor receptor-bound protein 2 (Grb2) adaptor protein (Okutani et al, 1994; Yamauchi et al, 1997), whereas phosphorylated Tyr1148 and Tyr1173 can recruit the Shc adaptor protein (Okabayashi et al, 1994). Recruitment and subsequent phosphorylation of Shc and/or Grb2 can lead to activation of the Erk MAPK pathway and thereby mediate activated cell proliferation or cell migration. Other studies have related specific EGFR tyrosine phosphorylation sites to cell phenotypic response, including those that have highlighted the role of Y992 and Y1045 in regulating EGFR trafficking (Helin & Beguinot, 1991; Helin et al, 1991; Sorkin et al, 1991, 1992). In addition to these well-characterized interactions, several pTyr sites on the receptor have been postulated to bind multiple adaptor or effector proteins,

[1]Department of Biological Engineering, Massachusetts Institute of Technology, Cambridge, MA, USA    [2]Koch Institute for Integrative Cancer Research, Massachusetts Institute of Technology, Cambridge, MA, USA    [3]Center for Precision Cancer Medicine, Massachusetts Institute of Technology, Cambridge, MA, USA    [4]Ludwig Institute for Cancer Research, La Jolla, CA, USA    [5]Moores Cancer Center, University of California at San Diego, La Jolla, CA, USA    [6]Department of Medicine, University of California at San Diego, La Jolla, CA, USA

Correspondence: fwhite@mit.edu

and several adaptors can bind to multiple sites. How these interactions are regulated in the cellular environment and how they control downstream signaling is still poorly understood.

Further complicating this picture, cytosolic tyrosine kinases and tyrosine phosphatases can be recruited to phosphorylated EGFR and function to provide positive and negative feedbacks to regulate receptor phosphorylation and activity. For instance, in a positive feedback loop, initial activation of EGFR can lead to activation of Src (or Src-family kinases) that can then phosphorylate Y845 in the EGFR activation loop, further increasing EGFR activity (Sato et al, 2003). Other cytosolic kinases such as the Abelson tyrosine kinase (Abl) have been shown to phosphorylate other cytosolic residues in the C-terminal tail, including Y1173 (Tanos & Pendergast, 2006). Tyrosine phosphatases, including PTP1B and SHP2, have been postulated to be recruited to the receptor and serve to dephosphorylate C-terminal tyrosine residues, thus serving as negative feedback to limit receptor activity (Östman et al, 2006). Inhibition of the phosphatases has been shown to have a strong impact on some autophosphorylation sites, even in the immediate-early stages after ligand stimulation (Reddy et al, 2016). Finally, crosstalk with other RTKs can add to the complexity, through heterodimerization or oligomerization and cross-phosphorylation of tyrosine sites on other RTKs or potentially mediated by activation of cytosolic tyrosine kinases (Vouri et al, 2016). Regardless of the mechanism, tyrosine phosphorylation of other RTKs may lead to adaptor recruitment, phosphorylation, and additional downstream signaling that could alter phenotypic outcome, including therapeutic resistance (Lynch et al, 2004). Because tyrosine phosphorylation controls many aspects of cellular and tumor biology, analysis of phosphorylation-mediated signaling networks can provide crucial information on novel biomarkers or resistance mechanisms. However, many approaches fail to provide predictive insights as the resulting models are based on few nodes with data collected from isolated interaction experiments (e.g., protein–protein interaction experiments) that do not consider the complexity of intracellular network interactions (Kovacs et al, 2015; Gill et al, 2017).

To attempt to elucidate the functional roles of individual EGFR C-terminal tyrosine phosphorylation sites in mediating cellular signaling and phenotypic outcome, NR6 cells lacking ErbB expression were infected to stably express WT EGFR or EGFR mutant isoforms containing one or more site-specific mutations of selected C-terminal tyrosine phosphorylation sites. Phosphoproteomic analysis of EGF-stimulated cellular signaling in each cell line demonstrated the resilience of the EGFR network to loss of selected pTyr sites, whereas also highlighting connections between particular pTyr sites and adaptor or effector proteins. To define associations between loss of selected pTyr sites, signaling network alterations, and cell biological effects, we quantified cell phenotypic responses, including receptor trafficking, cell proliferation, and cell migration. Intriguingly, despite a similar overall network response in each line, the phenotypic response to stimulation was, in many cases, significantly altered across cell lines expressing different mutant isoforms. To gain insight into the connection between our highly dynamic early timepoint phosphoproteomic dataset and the various quantitative phenotypic readouts, we developed a predictive data-driven model that highlights the importance of particular signaling nodes in mediating each phenotypic outcome. These data and the predictive model provide novel insights regarding EGFR signaling and suggest potential therapeutic targets to modulate cell biological response to EGFR stimulation.

# Results

## Development and initial characterization of EGFR-expression system

To evaluate the individual contributions of the C-terminal tyrosines in the EGFR signaling network response, mouse fibroblasts (NR6) lacking ErbB expression were retrovirally transfected to stably express human wtEGFR or mutant EGFR isoforms in which one or more tyrosines in the C-terminal tail were converted to a non-phosphorylatable phenylalanine residue (e.g., Y-to-F mutation) to evaluate loss of function. In total, nine cell lines were generated; one expressing wtEGFR, six containing single Y-to-F mutations (Y845F; Y992F; Y1045F; Y1068F; Y1148F; and Y173F), and two isoforms with most of the canonical C-terminal tyrosine phosphorylation sites mutated to phenylalanine. The latter two isoforms contain 5 Y-to-F mutations (Y992F, Y1068F, Y1086F, Y1148F, and Y1173F) (DY5), or 6 Y-to-F mutations (Y845F, Y992F, Y1068F, Y1086F, Y1148F, and Y1173F) (DY6) (Fig 1A). The DY5 and DY6 multiple mutant isoforms allow for evaluation of signaling effects in the context of loss of the major autophosphorylation sites; comparison of DY6 to DY5 also allows for assessment of gain-of-function of phosphorylation of Y845 located in the activation loop.

After successful transfection, expression levels were determined for all cell lines and replicates by flow cytometry (Fig S1A). By comparison with other cell lines with known levels of EGFR expression, we determined that wtEGFR was expressed at ~150,000 copies per cell in our transfected NR6 lines (Fig S1B). Most of the mutant isoforms were expressed at a similar level to wtEGFR, with the exception of Y845F and Y1045F, expressed at 0.5x and 3x of the levels of wtEGFR (Fig 1B). Phosphorylation of the Y1045 site has been associated with binding of the Cbl E3 ubiquitin ligase and subsequent ubiquitination of EGFR. Blocking phosphorylation at this site through Y-to-F mutation should therefore result in altered receptor trafficking, including increased recycling and decreased degradation, providing a potential explanation for the significantly greater cell surface expression of the Y1045F mutant isoform relative to wtEGFR.

After quantifying receptor surface expression, to assess the effect of each mutation on the cellular signaling network, cells were stimulated with 2 nM EGF for 0 s, 30 s, 1 min, 2 min or 5 min. At the appropriate time after stimulation, the cells were snap frozen in liquid nitrogen, lysed in cold 8M urea to preserve physiological signaling, and proteolytically digested to peptides. To accommodate the large number of samples (five time points per cell line, nine total cell lines, and three biological replicates per condition), peptides from each cell lysate were labeled with isobaric tandem mass tags and analyzed as 10-plex experiments (sample labeling scheme for each analysis in Fig S2) with pooled normalization controls. Labeled samples were subjected to two-step phosphotyrosine enrichment and subsequent liquid chromatography tandem mass spectrometry

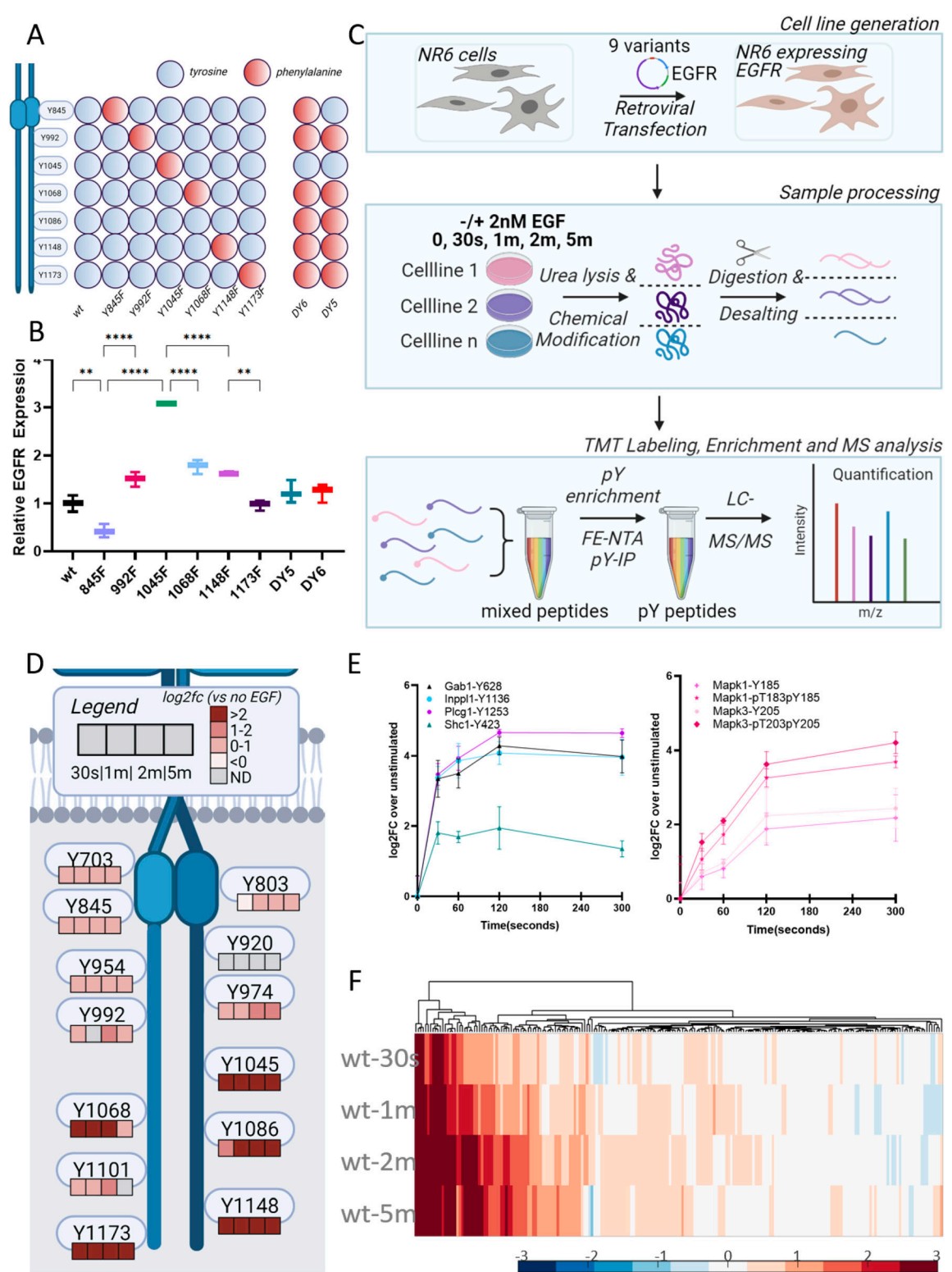

**Figure 1.  EGFR expressing model system exhibits strong EGF response.**
**(A)** A panel of nine cell lines that was generated, including WT, six single Y-to-F mutants, and two multiple mutant isoforms lacking 5 or 6 tyrosine residues. Red and blue dots indicate the presence of phenylalanine or tyrosine at residue location, respectively. **(B)** EGFR expression levels as determined by flow cytometry with EGF-labeled fluorophore, normalized to WT, data are presented as mean ± SD. *$P \leq 0.05$ (one-way ANOVA), n = 3. **(C)** Workflow schematic of NR6 retroviral transfection, cell stimulation, and sample processing, followed by TMT labeling and phosphotyrosine enrichment steps for LC–MS/MS analysis. **(D)** EGFR peptide phosphorylation levels in WT EGFR-expressing cells normalized to an unstimulated condition, $\log_2$ fold change, n = 3. **(E)** Temporal dynamics of phosphorylation levels of several downstream nodes in EGFR network, normalized to the unstimulated condition, $\log_2$ fold change. Data are presented as mean ± SD (t test) n = 3. **(F)** Hierarchical clustering of WT tyrosine phosphorylation data, normalized to the unstimulated condition, $\log_2$ fold change, data are presented as mean, n = 3.

(LC–MS/MS) analysis (Fig 1C). These analyses yielded quantitative data for 869 phosphotyrosine (pTyr) peptides, on average, for each analysis, with 254 pTyr sites being present across all conditions in at least two replicates and 217 pTyr sites present in all conditions and all replicates (quantitative pTyr data for all cell lines and conditions can be found in Table S1). Tyrosine phosphorylation sites were quantified in this study as most of the immediate-early signaling network downstream of EGFR activation (e.g., the EGFR network) is regulated by tyrosine phosphorylation, and we have previously shown that this pTyr-mediated network comprises hundreds of sites and is associated with downstream cell phenotypic responses. Additional insight may be gained by quantifying and modeling serine and threonine phosphorylation (pS/pT) in these systems, but much of this signaling lies downstream of the pTyr-mediated signaling network and most of the pS/pT sites are unaltered by EGF stimulation.

### NR6 cells expressing WT exhibit signaling network response to EGF stimulation

With this large dataset in hand, we initially interrogated pTyr temporal dynamics in wtEGFR-expressing NR6 cells as a starting point to ensure accurate signaling in the NR6 model system. Dynamic pTyr data from these cells displayed the expected behavior, with well-characterized autophosphorylation sites in the EGFR C-terminal tail (Y1045, Y1068, Y1086, Y1148, and Y1173) all displaying strong, significant increase in phosphorylation within 30 s after stimulation. Other sites on the receptor (Y703, Y803, Y845, Y954, Y992, Y1101) were detected and also demonstrated significantly increased phosphorylation levels upon EGF induction (Fig 1D). Although NR6 cells do not naturally express EGFR or other ErbB receptors, they are signaling competent, in agreement with multiple previous studies (Pruss & Herschman, 1977; Masui et al, 1991; Glading et al, 2000; Jamison et al, 2013). Signaling pathways downstream of the receptor in our wtEGFR-expressing NR6 cells are intact and responsive to EGFR stimulation, as shown by a strong temporal response of several well-characterized proximal adaptor protein phosphorylation sites and the ERK 1/2 mitogen-activated protein kinases (MAPK1 and MAPK3) (Fig 1E). Signaling effects of EGF stimulation are widespread in these cells, and beyond the canonical EGFR signaling network, we detected an overall increase in pTyr phosphorylation of the signaling network over time (Fig 1F) (Tong et al, 2014). Of the 217 pTyr sites detected in these cells, 59% (128/217) were significantly altered by EGF stimulation at one or more timepoints. As a negative control, parental, non-transfected NR6 cells were stimulated with EGF and subjected to the same analysis. Although an increase in phosphorylation was observed at the 30 s timepoint, the signaling network nodes responding to stimulation were largely comprised of stress-response signaling, including reactive-oxygen species-dependent activation of Src-family kinases along with JNK and p38, with no EGFR peptides detected in the analysis. Moreover, the overall trend indicates diminished activation at later timepoints, further suggestive of an EGFR-independent stress response (Fig S3). Taken together, these results suggest that NR6 cells transfected to express wtEGFR respond to EGF stimulation in a manner that is highly similar to EGFR signaling in other lines (Schneider et al, 1986), thus establishing this as a viable model system to study the effects of loss of phosphorylation on select C-terminal sites.

### EGFR network is largely resilient to mutation of C-terminal tyrosine residues

Having confirmed that NR6 cells transfected to express wtEGFR faithfully recapitulate EGFR signaling, we next interrogated the temporal pTyr phosphoproteomic data from NR6 cells expressing each of the mutant isoforms to assess the effect of each mutation on phosphorylation of EGFR, the adaptor proteins, and downstream signaling networks. To account for the different initial signaling states of each line (Fig S4), temporal phosphorylation data for each line were normalized to their own unstimulated ("0 s") time point, thus extracting quantitative changes induced by EGF stimulation at each timepoint. Interestingly, data from the mutant isoform-expressing NR6 lines indicate largely intact and robust signaling response to EGF stimulation, regardless of site-specific mutation. The overall resilience of EGFR signaling can be seen from the high degree of similarity (quantified in Fig S5) between the different cell lines in the heatmap in Fig 2A and the average network phosphorylation levels at each timepoint for each cell line in Fig 2B. Surprisingly, mutating the Y1148 residue did not seem to affect the overall network response compared with wtEGFR, even though this site is reported to be three to four times more phosphorylated than other sites on EGFR (Curran et al, 2015; Reddy et al, 2016). Moreover, even the cell lines expressing DY5 and DY6 isoforms, lacking 5 or 6 tyrosines, respectively, were able to induce a robust phosphorylation network response.

Correlation clustering of the EGFR phosphopeptide data confirmed high similarity in EGFR activation between isoforms (Fig S6). Drilling down further and comparing individual phosphopeptide levels between isoforms, the phosphorylation levels of Y1173 and Y1148 were unaffected by mutation of other tyrosine residues, whereas phosphorylation of Y954 was significantly down-regulated in isoform Y992F after 2 min of EGF treatment (Fig 2C and D). These data suggest that dominant phosphorylation sites such as Y1148 and Y1173 phosphorylation may be less dependent on the phosphorylation of other residues, whereas Y954 phosphorylation more directly correlates with EGFR's ability to phosphorylate Y992. Network nodes downstream of EGFR demonstrated robustness as well, as shown for selected well-characterized adapters and effectors of EGFR (Fig 2E). Slight but significant differences were observed between WT and Y1068F at the 2-min timepoint for the singly phosphorylated MAPK1 and MAPK3 and for Y1045F for the doubly phosphorylated variants. Despite similar phosphorylation response for most EGFR peptides, adaptor proteins, and the ERK MAPK effector proteins, decreased phosphorylation in response to stimulation was observed for multiple phosphorylation sites in the Y992F-mutant expressing cells and the Y845F-mutant-expressing cells (Fig 2F). These differences were recapitulated in clustering the isoforms based on the area under the curve (AUC) for all peptides, which revealed Y992F- and Y845F-mutant expressing cell lines among the least correlated with wtEGFR (Fig 2G). Co-correlation analysis of the individual timepoints further exposed the differences in response to EGF between these lines and wtEGFR (Fig 2H). This suggests that although the network appears to be highly resilient overall, selected mutations on the receptor can still alter network response to EGF stimulation.

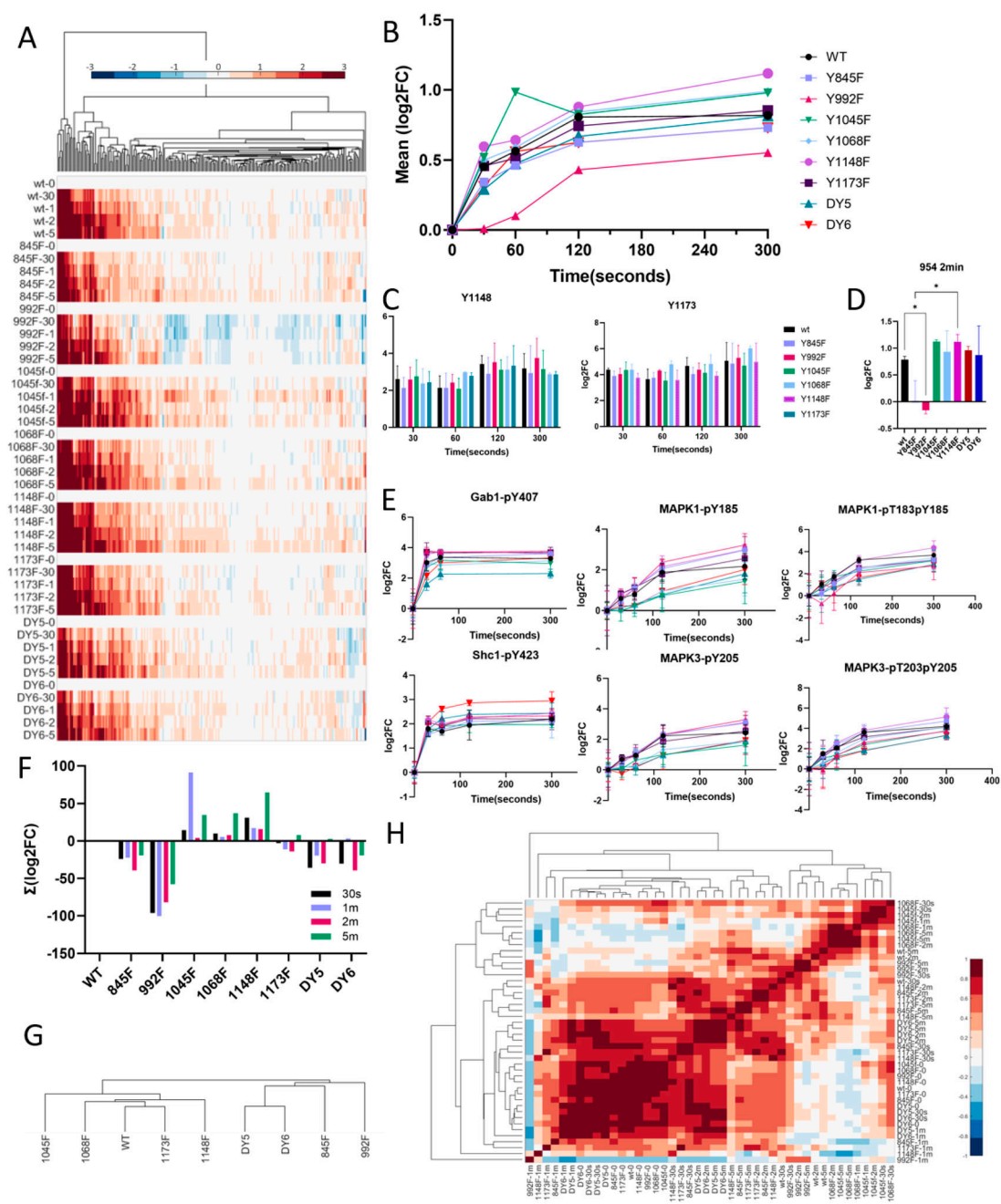

**Figure 2. EGFR network is largely resilient to loss of pY sites.**
**(A)** Hierarchical clustering of tyrosine phosphorylation data of all EGFR isoforms. Log$_2$ fold change over basal levels, data are presented as mean, n = 3. **(B)** Time course plot showing averaged log$_2$ fold change across peptides for each cell line at each timepoint, n = 3. **(C)** EGFR-Y1148 and EGFR-Y1173 phosphorylation levels in cell lines (if detected), log$_2$ fold change. No significant differences between conditions. Data are presented as mean ± SD (*t* test). **(D)** Tyrosine phosphorylation levels of EGFR peptides containing Y954, showing lack of EGF-induced phosphorylation in the Y992F isoform. Data are presented as mean ± SD, *P ≤ 0.05 (*t* test). **(E)** Time course plots of tyrosine phosphorylation levels of several downstream nodes of the EGFR network, normalized to unstimulated conditions, log$_2$ fold change. Data are presented as mean ± SD, *P ≤ 0.05 (*t* test), n = 3. **(F)** Sum of log$_2$ fold change for each cell line and timepoint demonstrating the overall amount of phosphorylation between timepoints and cell lines. **(G)** Dendrogram depicting how cell lines cluster based on similarity in EGF response, as calculated by area under the curve. **(H)** Co-clustering analysis of individual cell lines and timepoints demonstrate order of similarity between conditions. *P ≤ 0.05 (one-way ANOVA), n = 3.

## Downstream network analysis reveals network rewiring for Y992F isoform involving PLCγ-1 and other RTKs

To further interrogate signaling differences between mutant-isoform–expressing lines, we performed principal component analysis (PCA). Although most lines and conditions clustered are tightly together in the principal component (PC) space, stimulation timepoints from the Y992F-expressing lines occupied a distinct region, strengthening the observation that Y992F is an outlier in terms of EGF network response compared with the other cell lines

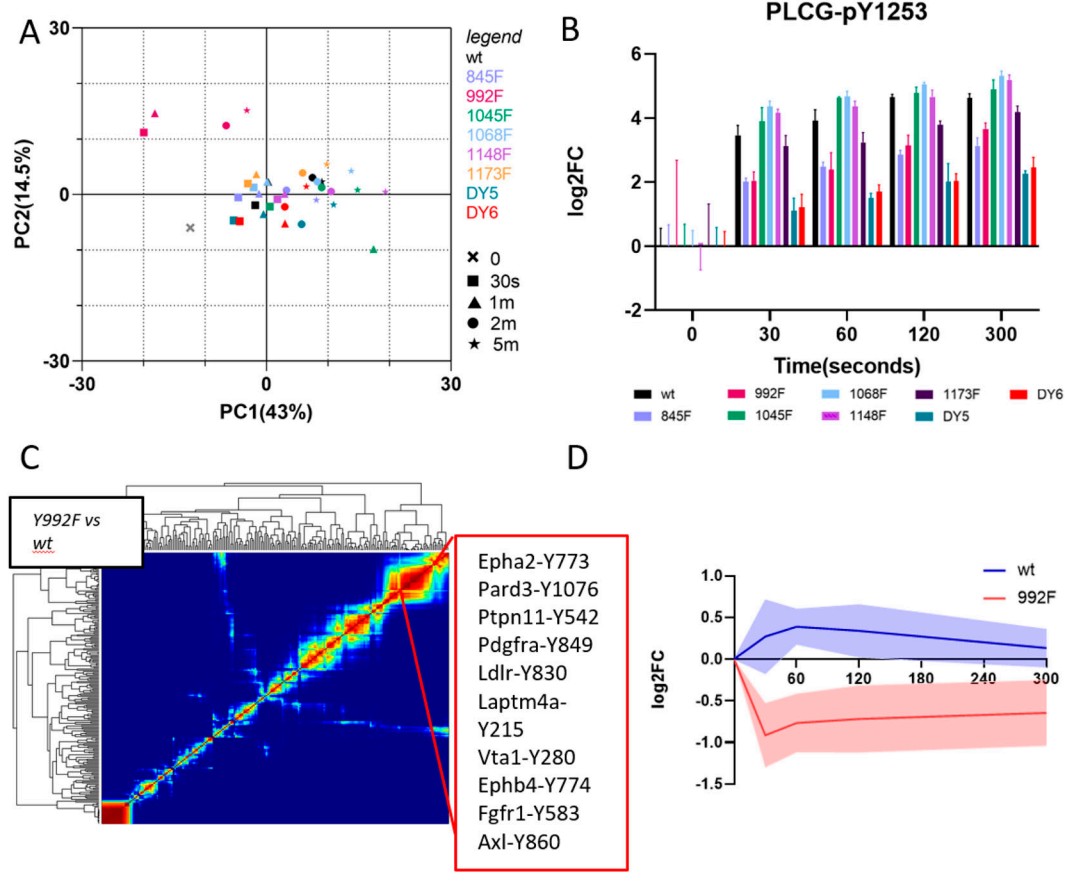

**Figure 3. Downstream network analysis reveals network rewiring for the Y992F isoform involving PLCγ and other RTKs.**
**(A)** Principle component analysis explaining 43% (PC1) and 14.5% (PC2) of the variance in the data, data are basal normalized. **(B)** Phosphorylation levels of PLCG-Y1253 at each timepoint per isoform. Log$_2$ fold change compared with basal. Data are presented as mean ± SD, n = 3. **(C)** Self-organizing map clustering to identify similarly differentially regulated peptides between wt and Y992F mutant isoform. Neural network iterations = 1,000. **(C, D)** 95% confidence interval of peptides from the highlighted cluster in panel (C), depicting difference in trend between EGFR isoforms. Data presented are mean log$_2$ fold changes and 95% confidence interval, n = 3.

(Fig 3A). Within the PC space, PC1 largely explains the time component, whereas PC2 appears to depend on the level of phosphorylation in response to EGF (e.g., Y1045F, Y1148F, and Y1068F have a stronger overall phosphorylation response compared with Y992F and the multiple mutants).

Consistent with the documented role of phosphorylated Y992 in PLCγ-1 binding and activation (Rotin et al, 1992), PLCγ-1 phosphorylation was significantly decreased in the Y992F isoform and in DY5 and DY6, both of which contain the Y992F mutation, relative to wtEGFR across all timepoints, confirming this site as an important node for PLCγ-1 activation (Fig 3B). PLCγ-1 phosphorylation was down in the Y845F isoform as well. To gain further insight into the earlier observation of decreased phosphorylation on multiple nodes in the Y992F line in the hierarchical clustering analysis (Fig 2A), we generated a SOM to identify clusters of co-regulated peptides that were differentially regulated in Y992F compared with wtEGFR (Fig 3C). SOMs use neural networks to display similarities in the data, thereby revealing clusters of peptides that are similarly differentially regulated between isoform lines. In the SOM comparing Y992F with wtEGFR signaling response, one of the large clusters displayed an increase in phosphorylation upon EGF

treatment at 30 s in the wtEGFR-expressing line (blue line), whereas these same peptides decrease in phosphorylation upon EGF stimulation in the Y992F-expressing line (red line) (Fig 3D). Interestingly, most of the peptides in this cluster belonged to other RTKs, including PDGFRα, EPHA2, LDLR, AXL, FGFR1, and EPHB4. These data suggest a potential regulatory function for pY992 in modulating crosstalk with other RTKs or by stabilizing otherwise autoinhibitory intramolecular interactions (Lemmon et al, 2014). Given the role of PLCγ-1 as a negative feedback regulator for homodimer formation (Thirukkumaran et al, 2020), the increased availability of PLCγ-1 because of lack of binding opportunity at the 992 residues in the Y992F isoform could have inhibited dimer formation and hence reduced phosphorylation of other RTKs.

## Phenotypic characterization confirms known biology and reveals interesting trends

As the phosphoproteomic data reflect overall signaling network similarity and some mutant-specific differences, our findings raised the question as to whether and how this variation translates to cellular response. To evaluate the phenotypic consequences of

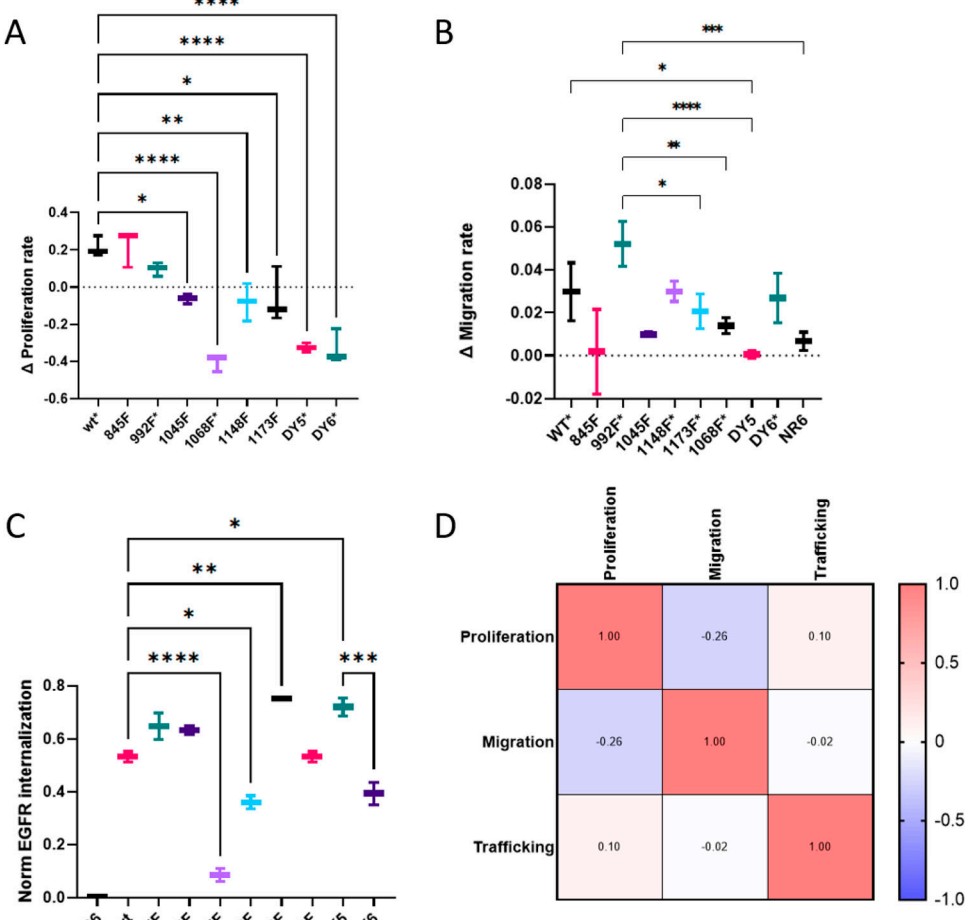

**Figure 4. EGFR mutants demonstrate different phenotypic response to EGF.**
**(A)** Proliferation rates measured by confluency % over 24 h, Data are presented as mean ± SD. *$P ≤ 0.05$ (one-way ANOVA), n = 3. * indicates significant difference compared with wt. * at label indicates significant difference between EGF and no EGF condition. **(B)** Migration rates determined by measuring scratch wound assay closing rate over 24 h. Data are presented as mean ± SD. *$P ≤ 0.05$ (one-way ANOVA), n = 3. * indicates significant difference. * at label indicates significant difference between EGF and no EGF condition. **(C)** Receptor trafficking as determined by flow cytometry and pHrodo-EGF conjugate. Untransfected NR6 were used as negative control and normalizing channel. Data are presented as mean ± SD. *$P ≤ 0.05$ (One-way ANOVA), n = 3. **(D)** Correlation analysis between phenotypes, data presented are Pearson correlation coefficients. n = 3.

tyrosine loss-of-function, proliferation, migration, and internalization rates were measured (Fig 4). Confluency measurements taken over time for each line in the absence or presence of EGF were compared to quantify an EGF-response proliferation rate. EGF-driven proliferation rates were significantly lower for 6 out of 8 mutant isoforms compared with wtEGFR (Fig 4A), with 3 of these lines yielding a counterintuitive decrease in proliferation in the presence of EGF as compared with the absence of EGF. This result appears to be linked to Y1068F (and may be related to dynamics of the singly phosphorylated ERK activation loop [Fig 2E]) as all three lines contain this mutation. To evaluate EGF-stimulated cell migration rates, temporal dynamics of wound healing were quantified in confluent plates for each cell line, in the presence or absence of EGF. EGF stimulated cell migration at a similar rate for most of the mutant isoform expressing lines as compared with wtEGFR, with only DY5 being significantly different (Fig 4B). In comparing the mutant lines with each other, Y992F-expressing cells were significantly more responsive to EGF-stimulated migration as compared with most other lines, despite decreased phosphorylation on many signaling nodes. In addition to proliferation and migration as measures of downstream phenotypes, receptor trafficking to endosomes was evaluated as a more proximal phenotype. The pHrodo-EGF conjugate demonstrates increased fluorescence in

acidic environments. We used this conjugate to quantify internalization and localization of the EGF–EGFR-bound complex to low-pH endosomes. The results of this assay highlight significant differences in trafficking between wtEGFR and 4 mutant isoforms (Fig 4C). Perhaps not surprisingly, minimal endosomal trafficking was detected for Y1045F, in accordance with its reported role in CBL binding, ubiquitination, and receptor trafficking. Y1068F also showed decreased endosomal trafficking compared with WT. This effect may be explained by Y1068's function as a Grb2-binding site, as Grb2 is reported to be important in early stages of endocytosis where EGF-activated receptors are recruited into clathrin-coated pits (Jiang et al, 2003). Interestingly, DY6 also showed decreased trafficking, however, when Y845 function was recovered (DY5), trafficking was at least as high as WT levels, although both isoforms still contained the Y1068F mutation (amongst others), which by itself caused decreased trafficking. Correlation analysis of the different phenotypes revealed a slight negative correlation between migration and proliferation, which supports the principle of phenotype-switching as cells decide which phenotype to pursue (Fig 4D) (Li & Xie, 2011). Curiously, there appears to be minimal association between network-level phosphorylation and phenotypic outcome in response to EGFR stimulation, as most mutant isoforms displayed relatively minor signaling changes at the network level relative to wtEGFR, yet several had significantly altered

phenotypic response. On the other side, cells containing the Y992F mutant isoform were the most different from wtEGFR in signaling, yet, this same mutation appeared to have a minimal phenotypic impact.

### Partial least squares regression model reveals Eps8- and PLCγ-1-mediated mechanisms associated with cellular phenotype

To gain greater insight into the associations between protein site phosphorylation dynamics and phenotypic response to EGF stimulation, we used partial least square regression (PLSR) to integrate these quantitative datasets and determine drivers of the observed phenotypic differences (Wolf-Yadlin et al, 2006; Kumar et al, 2007; Johnson et al, 2013). After selecting the optimal number of latent variables, the model was generated based on leave-one-out cross validation (LOOCV). The model was predictive of phenotype with a fit to data average $R^2$ of 0.97, whereas cross-validation provided highly predictive $Q^2$ values > 0.75 for all phenotypes and signaling data timepoints (Table S3), and using the area under the curve as a metric of total phosphorylation for each site over the time course. To assess the impact of Y845F and Y1045F cell lines on the model, we removed the data from these lines, individually or in combination, and generated a PLSR model from the reduced dataset. These new models featured a similar fit to data $R^2$ and predictive $Q^2$ values, suggesting that Y845F and Y1045F were not adversely affecting model accuracy (Table S3). From this model, variable importance in projection scores were calculated to identify the most critical contributors to the PLSR model.

To assess the accuracy of the predicted model associations, we checked the phosphorylation sites predicted to be positively associated with cell proliferation (Table S2). In agreement with previous reports, phosphorylation of the activation loop of the ERK MAPKs (MAPK1 and MAPK3) at 2 and 5 min after stimulation was found to be positively associated with proliferation, as was the phosphorylation of SCL38A2 (Y41) (Wei & Liu, 2002; Eswarakumar et al, 2005; Wolf-Yadlin et al, 2006). In addition, CDK1/2 Y15 is positively associated with cell proliferation in our model; although this site is known to inhibit progression through the cell cycle, cells that proliferate more frequently have increased CDK Y15 phosphorylation, agreeing with the model prediction. On the other side, tyrosine phosphorylation of Afadin (Y1230, Y1285), Girdin (Ccdc88a) (Y1801), and Erbin (Y1097) are negatively associated with proliferation. Tyrosine phosphorylation of these proteins has been shown to promote cell migration, further suggesting a cell decision process downstream of EGFR, where phosphorylation of migration-associated proteins is negatively correlated with cell proliferation (De Donatis et al, 2008; Stallaert et al, 2018). For the PLSR model integrating protein phosphorylation with cell migration, the above sites on Afadin, Girdin, and Erbin were all positively associated with cell migration, as were sites on delta-catenin (Y228, Y865, Y904) and PLC γ-1 (Y771, Y1253), also supporting the predictive power of this model. The PLSR model integrating protein phosphorylation and receptor endocytosis has similar predictive power, with Q2 = 0.87, yet the signaling networks associated with receptor internalization and trafficking are less characterized, so interpretation of the model predictions is more challenging. For instance, phosphorylation of Grb2-associated binding protein 1 (Gab1) on Y628 and Y660 and phosphorylation of the Erk 1/2 MAPK activation loops (T183/Y185; T203/Y205) are positively associated with receptor endocytosis, in agreement with a previous publication suggesting that Gab1 is recruited to early endosomes and is the primary mediator of Erk MAPK signaling after receptor endocytosis (Kostenko et al, 2006). However, the roles of other sites in this phenotype, including a strong negative association between EGFR internalization and PLCγ-1 phosphorylation (Y771, Y783, Y1253), are less clear. It may be that EGFR phosphorylation of PLCγ-1 occurs on the cell membrane, and therefore, that increased internalization of EGFR leads to decreased phosphorylation of PLCγ-1, but this hypothesis would need to be shown experimentally.

One of the strongest contributors to the proliferation and migration models is pY601 of EGFR pathway substrate 8 (Eps8) (Fig 5A and B). Eps8 is a substrate of EGFR and has been shown to be involved in regulating cancer progression (Fazioli et al, 1993; Thiel & Carpenter, 2007; Li et al, 2019). In our PLSR models, Eps8 was found to be negatively correlated with proliferation, whereas it was positively correlated with migration. In agreement with our model predictions, phosphorylation of Eps8 Y601 has been reported to block proliferation and to promote cell migration (Cattaneo et al, 2012; Ding et al, 2013). To gain further insight into protein phosphorylation sites that may be co-regulated with Eps8, we performed co-correlation clustering. This analysis showed that Eps8 was most closely correlated with Y104 of the DEAD box helicase DDX3X, a protein whose phosphorylation has not been previously linked to the EGFR signaling pathway, yet DDX3X has been recently suggested to play a role in EGFR tyrosine kinase inhibitor resistance (Nozaki et al, 2014). Also, highly co-regulated with Eps8 are Girdin, Filamin b, RNA-binding motif protein 3 (Rbm3), TRIO and F-actin binding protein, and Grb2, all of which were positively correlated with cell migration and most of which were negatively associated with cell proliferation (Fig 5C). Intriguingly, several of these proteins, including DDX3X, Rbm3, and TRIO and F-actin-binding protein, have not been previously linked to EGFR signaling, yet through our analysis of signaling networks associated with different mutant isoforms of EGFR, we were able to not only connect these protein phosphorylation sites to EGFR activity, but also suggest associations with phenotypic outcomes downstream of EGFR activation.

## Discussion

Given the well-established role of EGFR as a driver oncogenic kinase in many cancer types, coupled with the need to improve EGFR-targeting strategies in the clinic and the dependence on phosphorylation of C-terminal tyrosines for signal propagation, we took a mechanistic, phosphoproteomics-based approach to investigate the contribution of individual EGFR C-terminal tail tyrosines to the EGFR signaling network. In this work, we used NR6 murine fibroblasts that lack endogenous ErbB expression as a model system. Although NR6 cells do not naturally express EGFR or other ErbB receptors, they are signaling competent, and NR6 cells transfected with human EGFR have been shown to respond to EGF stimulation by increasing proliferation and migration, suggesting intact downstream signaling

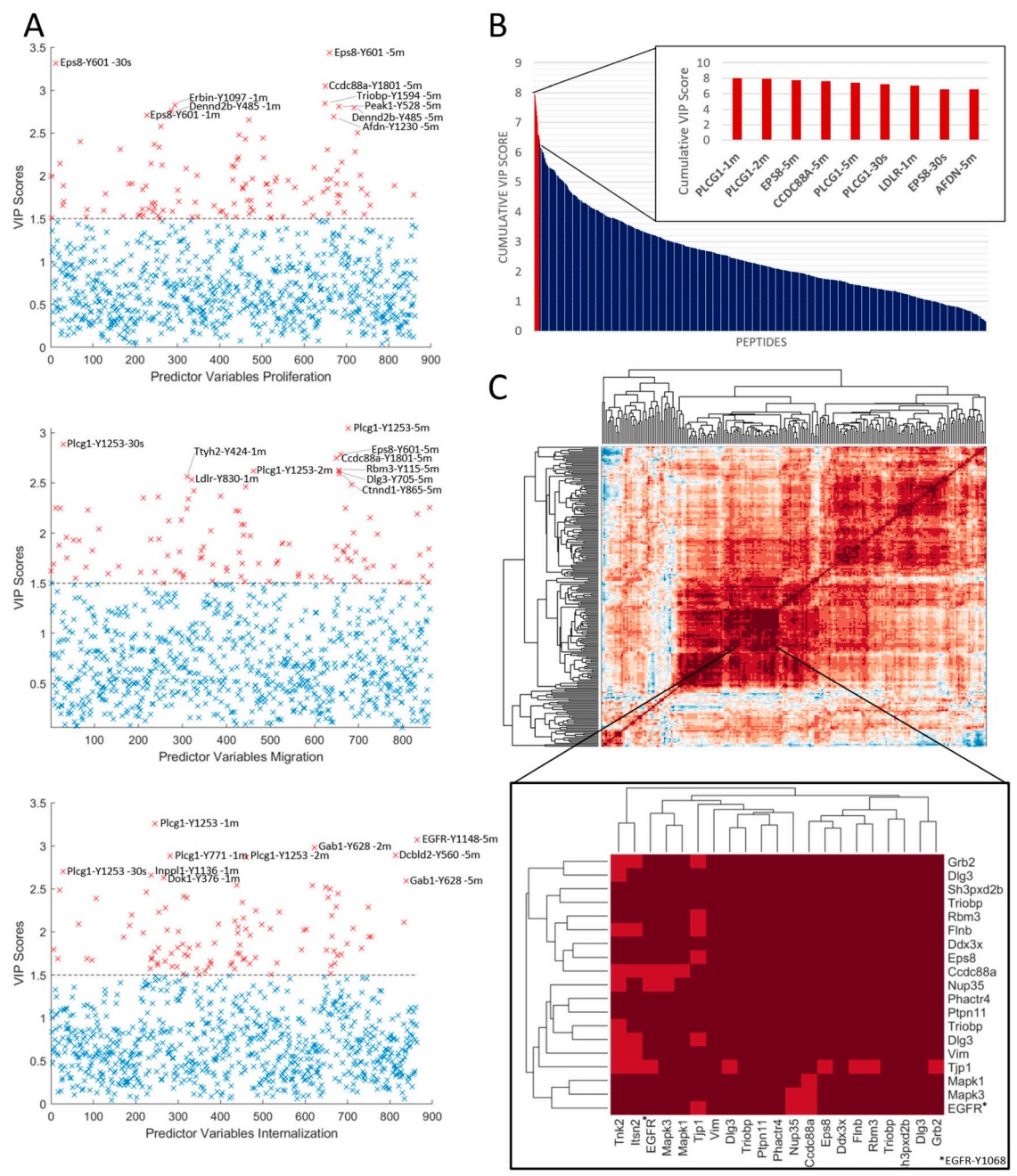

**Figure 5. PLSR reveals PLCγ1- and Eps8-mediated mechanisms predictive of cell phenotypes.**
**(A)** Variable importance in the projection statistics depicting most predictive nodes for each phenotype based on signaling data (AUC). Data presented based on three-component PLSR, cross-validated model. **(B)** Cumulative variable importance in the projection scores across models, rank ordered. **(C)** Co-correlation map of basal-normalized pTyr data (mean log$_2$ fold change) highlighting subcluster containing Eps8 peptide (Y602). PLSR model built using LOOCV and three components, R2, and prediction accuracy (Q2) values can be found in Table S3.

networks and compatibility of murine cells to expression of human EGFR in agreement with multiple previous studies (Pruss & Herschman, 1977; Masui et al, 1991; Glading et al, 2000; Jamison et al, 2013). Moreover, mouse adapter proteins appear to recognize human EGFR as there are minimal differences in the TKD and C-terminal sequence of EGFR between humans and mice. In addition, there are virtually no differences in the amino acid sequences directly surrounding the tyrosine sites on the intracellular domain.

Here, we generated individual and combined Y>F mutant isoforms and stably expressed each of these isoforms in NR6 cells to evaluate loss of function effects for selected C-terminal tyrosine phosphorylation sites.

Basal activity of the stably expressed EGFR isoforms in our NR6 cells resulted in a slightly altered baseline signaling state for each isoform-expressing line. Although we were able to correct for this baseline signaling by normalizing the temporal data to the non-

stimulated condition (e.g., the "zero" timepoint), other model system choices may provide for a more straightforward comparison. For instance, stable expression allows the cells to adapt/rewire to each mutant EGFR isoform. One could opt for a transient expression system to limit the amount of rewiring in the cells before stimulation, although the dynamic nature of transient expression, coupled with the difficulty of obtaining consistency in EGFR expression levels across replicates and lines, would be highly challenging. CRISPR-based gene editing to mutate selected tyrosine phosphorylation sites in endogenously expressed EGFR would also enable interrogation of loss-of-function of each site without exogenous stable expression. However, our attempts to achieve gene editing of EGFR sites in several epithelial lines were not successful, potentially because of the inefficiency of current gene editing approaches, making it highly challenging to achieve editing in both genomic copies of EGFR. Moreover, many epithelial cells are dependent on EGFR expression, and thus mutation of one or both copies may have been negatively selected.

Given some of the established connections between adaptor proteins and selected EGFR C-terminal tail phosphorylation sites, our initial hypothesis at the start of the project was that loss of a given phosphorylation site might lead to loss of signal for one or more adaptor proteins and associated signaling pathways. However, to our surprise, the EGFR signaling network was highly resilient, and signaling network response was highly conserved for each loss-of-function mutant isoform, even those with a combined mutation of 5 or 6 tyrosines. The strong similarity in network response to EGF stimulation regardless of loss of tyrosine function may be associated with the ability of most of the SH2 and phosphotyrosine binding domains on adapter proteins to bind to multiple pTyr sites in the C-terminal tail of the receptor. In the context of loss of one or more sites, these domains may bind to a less favorable site and thus preserve protein–protein interactions and the cellular signaling response. Notably, not all adapter protein domains bind to multiple sites; Cbl appears to have strong preference for Y1045 and PCLg appears to have strong preference for Y992, as indicated by our results and by multiple previous studies. It is also possible that the non-native environment of NR6 cells may regularize the signaling network response across different mutant isoforms. Unfortunately, the ability of the signaling network to compensate for loss of one or more tyrosines made it challenging to isolate the functional interactions of a given tyrosine site.

Previous work has connected signaling network nodes in the ErbB network with consequent phenotypic response to EGFR or Her3 stimulation (Wolf-Yadlin et al, 2006). We used a similar modeling framework to generate integrative computational models focused on three phenotypes that are known to be regulated by EGFR and cover various degrees of cellular response. Proliferation and migration occur on the hours-to-days timescale, whereas receptor internalization is an earlier phenotypic response that could be measured on the minute timescale. Using quantitative data for hundreds of tyrosine phosphorylation sites across multiple time points per cell line and nine different cell lines, we were able to generate robust and highly predictive models for each phenotype. Although cellular responses such as migration and proliferation take hours-to-days to occur, phosphorylation changes in the immediate-early timepoints, as early as 30 s or 1 min, were found to

be highly important in predictive models for these longer-term responses, in agreement with previous models of HER-family signaling networks (Wolf-Yadlin et al, 2006; Kumar et al, 2007; Johnson et al, 2013). These model predictions highlight the importance of measuring early phosphorylation events in the signaling network, as these events effectively set the course for the future cellular response to stimulation.

Phosphorylation of PLCγ-1 and Eps8 emerged as some of the most important sites in the models, with Eps8 as a major feature negatively associated with cell proliferation and positively associated with cell migration. Among the EGFR phosphorylation sites, Eps8 phosphorylation was strongly correlated with phosphorylation levels of the 1,068 residues (Fig 5C). Eps8 is known to promote cell migration and also emerged as a strong positive factor in this model, suggesting that phosphorylation of this protein may be part of the switch mechanism used by the cell for phenotype decision-making (Cattaneo et al, 2012; Ding et al, 2013). Small molecule inhibitors targeting the EGFR/Eps8 complex in NSCLC and breast cancer models have shown promising results, supporting Eps8 as a potential therapeutic target in cancer (Li et al, 2019). PLCγ-1 also emerged as one of the strongest contributors to the models, with a positive association in the cell migration model and a negative association in receptor endocytosis. Although the role of PLCγ-1 in promoting cell migration downstream of receptor activation is well established, the connection between receptor endocytosis and PLCγ-1 requires further investigation. Regression-based models provide associations, but often fail to provide a mechanism. In this case, it is not obvious whether increased receptor endocytosis leads to decreased PLCγ-1 phosphorylation or whether decreased PLCγ-1 phosphorylation leads to increased receptor trafficking, potentially through altered cAMP and Ca$^{2+}$ signaling. In addition to Eps8 and PLCγ-1, our regression models also highlight the role of other protein phosphorylation sites in mediating the cell response decision between migration and proliferation. For instance, Girdin (CCdc88a, also known as GIV) emerged as a highly important protein in the proliferation and migration models. Several manuscripts over the past few years have suggested that Girdin may serve to connect RTKs such as EGFR to G-protein-coupled receptors to thereby modulate cAMP levels and control the proliferation/migration cell decision process (Lin et al, 2014; Bhandari et al, 2015; Getz et al, 2019). Our data suggest that Y1801 on Girdin may play an important role in this process, as phosphorylation of this site was negatively associated with cell proliferation and positively associated with cell migration. Similarly, phosphorylation of RBM3 Y115 also emerged as a strong predictor in the proliferation and migration models. Although this protein has been previously linked to cell migration, in agreement with our model prediction, RBM3 has not been associated with EGFR signaling previously and Y115 appears to be a novel phosphorylation site. Additional studies are needed to characterize the role of this phosphorylation site on RBM3 function and to determine the mechanism underlying association with cell migration.

This mechanistic approach to uncover novel insights into the EGFR signaling network is easily translatable to other RTKs. It would be interesting to see if all RTKs have similar resilience in signaling response to loss-of-function mutation to one or more tyrosines or whether there is a hierarchy of sites on selected RTKs. Using

regression models to couple quantitative pTyr phosphoproteomics with quantitative measurements of cell biological response resulted in the correlation of poorly characterized phosphorylation sites with particular cell responses, highlighting potential functional roles for these sites. Application of this same approach to less-studied RTKs should provide a plethora of novel associations that should lead to much better understanding of these signaling networks. Previous work utilizing cell and molecular biological approaches has provided connections between selected phosphorylation sites and downstream cell phenotypic response to stimulation (Helin & Beguinot, 1991; Helin et al, 1991; Sorkin et al, 1991, 1992). These studies required a priori knowledge of the phosphorylation sites to be interrogated and had to be performed through mutagenesis of each phosphorylation site. Here, through proteomics and computational modeling, we were able to generate predicted associations for many phosphorylation sites to each phenotype. Follow-on studies to interrogate the mechanistic connections for each of these predicted associations are still required and are likely to yield novel insights into signaling network regulation underlying cell decision processes after receptor stimulation.

# Materials and Methods

### Plasmid preparation and extraction

Seven out of nine plasmid constructs were kindly donated by F Furnari Laboratory (UCSD). WT and 1045F mutant plasmids were generated from 845F plasmid using Genscript site-directed mutagenesis service and sequences verified with Sanger sequencing (EtonBio). Plasmids were pLNCX2 retroviral vectors with a neomycin and ampicillin-resistance gene, and the EGFR coding sequence containing a variety of Y-to-F mutations (Y845F, Y992F, Y1045F, Y1068F, Y1148F, Y1173F, DY6) (6 Y-to-F mutations [Y845F,Y992F, Y1068F,Y1086F,Y1148F,Y1173F]), DY5 (DY6 except F845Y) and WT plasmid DNA was transformed into DH5$\alpha$ *E. coli* (NEB 5-alpha competent E. Coli [High Efficiency]) according to NEB protocol. Cells from a −80°C freezer were thawed on ice for 5 min, and 5 $\mu$l of plasmid DNA was added to the cell mixture. The mixture was incubated on ice for 30 min and then heat shocked in a water bath at 42°C for 45 s. The mixture was placed on ice for 5 min 950 $\mu$l of super optimal broth with catabolite expression (SOC) medium was added to the mixture, which was then shaken at 37°C for 60 min. Subsequently, 100 $\mu$l was spread onto a Luria Broth (LB) agar with an ampicillin plate and incubated overnight at 37°C. Individual colonies were picked and grown for ~12 h at 37°C in 10 ml of LB media with ampicillin (100 mg/ml, 1,000x). Culture was spun down at 4°C and 9,500$g$ for 10 min. Plasmid DNA was isolated using the QIAGEN Spin Miniprep Kit. NanoDrop (Thermo Fisher Scientific) was used to determine DNA concentration.

### Generation of retrovirus

On day 0, 2 × 10$^5$ 293T cells/well were plated in six well-plates. The next day, 4 $\mu$l of fuGENE HD Transfection Reagent (E2311; Promega) was added to 100 $\mu$l of Opti-MEM (Gibco) and vortexed to mix and incubate for 5 min at room temperature (RT). Envelope protein VSV-G and packaging genes Gag/Pol were combined with plasmid DNA; 333, 666, and 500 ng, respectively. The DNA mix was added dropwise to the solution containing fuGENE and HD transfection reagent, mixed, and left to incubate at RT for 30 min. The resulting mixture was then added dropwise to the 293T cells. On day 2, the media were replaced with regular DMEM containing 1% pen/strep and 10% FBS using a P1000 to minimally disturb the cells. NR6 cells were plated in six well-plates at 5 × 10$^4$ cells/well. The following day, virus was harvested from the 293T cells by collecting media and passing it through a .45 $\mu$m filter. Any virus not immediately used for transduction was stored at −80°C.

### Retroviral transduction

Polybrene (Sigma-Aldrich) was added to the solution at 6 $\mu$g/ml, and the media for NR6 cells were replaced with the filtered DMEM containing the virus/polybrene mixture. 8 h later, the media were removed and DMEM supplemented with 10% FBS and 1% penicillin/ streptavidin (P/S) added onto the cells. On day 6, Geneticin (G418; Gibco) was added at 45 $\mu$l/well. One day later, the cells were split and kept on G418 selection pressure.

### Flow cytometry for EGFR expression and endocytosis

To select successfully transfected cells and to ensure comparable expression levels between cell lines and replicates, EGFR expression was determined by flow cytometry. Biotinylated EGF, complexed to Alexa Fluor 488 (E13345; Invitrogen) was added to each NR6 cell line at a concentration of 0.5 $\mu$g/ml and incubated at 37°C for 5 min. Cells were put on ice, spun down at 4°C at 1,200$g$ for 5 min, and washed twice with cold PBS before analysis on the BD FACS-canto Clinical Flow Cytometry System. When selecting successfully transfected cells, the BD FACSAria Sorter was used and gating for EGFR-expressing cells was determined with respect to a negative control.

To evaluate EGFR internalization and trafficking, pHrodo EGF 488 conjugate (P35375; Thermo Fisher Scientific) was added to each NR6 cell line at a concentration of 0.5 $\mu$g/ml and incubated at 37°C for 5 min. Cells were put on ice, collected from the plate using Accutase, spun down at 4°C at 1,200$g$ for 5 min, and washed twice with cold PBS before analysis. Data were acquired using the BD FACSDIVA Software and further analyzed using FlowJo.

### Cell culture and EGF stimulation

NR6 mouse fibroblasts (gifted by F Furnari Laboratory; UCSD) were maintained in DMEM (Corning) medium supplemented with 1% penicillin–streptomycin (5,000 U/ml, Gibco), 2% G418 (Gibco), and 10% FBS (Gibco). ~2 × 10$^5$ cells were seeded and incubated for 2 d to reach ~80% confluency in 10 cm dishes. For each line and each biological replicate, one plate of cells was used to determine EGFR expression level using Alexa Fluor 488-conjugated EGF and flow cytometry, as described above. Other plates of cells (one 10 cm plate/time point) were stimulated with 2 nM EGF (Peprotech) in

serum-free media for 30 s, 1 min, 2 min or 5 min or left untreated as a control (0 min timepoint).

## Sample preparation for MS analysis

After EGF stimulation, the media were promptly discarded and cells were snap frozen on liquid N2 for instant arrest of all signaling events. Cells were lysed on ice using 500 $\mu$l 8M urea (Sigma-Aldrich) per 10 cm plate. A bicinchoninic acid (BCA) protein concentration assay (Pierce) was performed according to the manufacturer's protocol to estimate the protein concentration in each lysate. Cell lysates were reduced with 10 mM DTT for 1 h at 56°C, alkylated with 55 mM iodoacetamide for 1 h at RT shielded from light, and diluted fivefold with 100 mM ammonium acetate, pH 8.9, before trypsin (Promega) was added (20:1 protein:enzyme ratio) for overnight digestion at RT. The resulting solutions were acidified with 1 ml of acetic acid (HOAc) and loaded onto C18 Sep-Pak Plus Cartridges (Waters), rinsed with 10 ml of 0.1% HOAc, and eluted with 10 ml of 40% Acetonitrile (MeCN)/0.1% HOAc. Peptides were divided into XYZ microgram aliquots, and sample volume was reduced using a vacuum centrifuge (Thermo Fisher Scientific) and then lyophilized to dryness for storage at −80°C.

TMT labeling for multiplexed analysis was performed according to the manufacturer's protocol. Samples, each containing ~200 $\mu$g peptides, were resuspended in 35 $\mu$l HEPES (pH 8.5), vortexed, and spun down at 17,000$g$ for 1 min. 400 $\mu$g of a given channel of TMT10plex (Thermo Fisher Scientific) in anhydrous MeCN was added per sample. Samples were shaken at 15$g$ for 1 h, after which the labeling reaction was quenched using 5% hydroxylamine (50%; Thermo Fisher Scientific). After another 15 min on the shaker, all samples were combined using the same pipette tip to reduce sample loss, and sample aliquots were washed twice with 40 $\mu$l 25% MeCN/0.1% HOAc which was added to the collection tube to improve yield. Sample volume was reduced using a vacuum centrifuge and then lyophilized to dryness for storage at −80°C.

## Phosphopeptide enrichment

Immunoprecipitation (IP) and using immobilized metal affinity chromatography were used sequentially to enrich samples for phosphotyrosine-containing peptides. TMT-labeled samples were incubated in IP buffer consisting of 1% Nonidet P-40 with protein G agarose beads conjugated to 24 $\mu$g of 4G10 V312 IgG and 6 $\mu$g of PT-66 (P3300; Sigma-Aldrich) overnight at 4°C. Peptides were eluted with 25 $\mu$l of 0.2% trifluoroacetic acid for 10 min at room temperature; this elution was performed twice to improve yield. Eluted peptides were subjected to phosphopeptide enrichment using immobilized metal affinity chromatography-based Fe-NTA spin column to reduce nonspecific, non-phosphorylated peptide background. High-Select Fe-NTA enrichment kit (Pierce) was used according to the manufacturer's instructions with following modifications. Eluted peptides from IP were incubated with Fe-NTA beads containing 25 $\mu$l binding washing buffer for 30 min. Peptides were eluted twice with 20 ml of elution buffer into a 1.7 ml microcentrifuge tube. Eluates were concentrated in speed-vac until ~1 $\mu$l of the sample remained, and then resuspended in 10 $\mu$l of 5% acetonitrile in 0.1% formic acid. Samples were loaded directly onto

an in-house constructed fused silica capillary column (50 micron inner diameter [ID] × 10 cm) packed with 5 $\mu$m C18 beads (YMC gel, ODS-AQ, AQ12S05) and with an integrated electrospray ionization tip (~2 micron tip ID).

## LC–MS/MS analysis

LC–MS/MS of pTyr peptides were carried out on an Agilent 1260 LC coupled to a Q Exactive HF-X mass spectrometer (Thermo Fisher Scientific). Peptides were separated using a 140-min gradient with 70% acetonitrile in 0.2 mol/l acetic acid at a flow rate of 0.2 ml/minute with an approximate split flow of 20 nl/minute. The mass spectrometer was operated in data-dependent acquisition with the following settings for MS1 scans: m/z range: 350–2,000; resolution: 60,000; AGC target: $3 \times 10^6$; maximum injection time (maxIT): 50 ms. The top 15 abundant ions were isolated and fragmented by higher energy collision dissociation with the following settings: resolution: 60,000; AGC target: $1 \times 10^5$; maxIT: 350 ms; isolation width: 0.4 m/z, collisional energy (CE): 33%, dynamic exclusion: 20 s. Crude peptide analysis was performed on a Q Exactive Plus mass spectrometer to correct for small variations in peptide loadings for each of the TMT channels. Approximately 30 ng of the supernatant from pTyr IP was loaded onto an in-house-packed precolumn (100 $\mu$m ID × 10 cm) packed with 10 mm C18 beads (YMC gel, ODS-A, AA12S11) and analyzed with a 70-min LC gradient. MS1 scans were performed at the following settings: m/z range: 350–2,000; resolution: 70,000; AGC target: $3 \times 10^6$; maxIT: 50 ms. The top 10 abundant ions were isolated and fragmented with CE of 33% at a resolution of 35,000.

## Peptide identification/quantification

Mass spectra were processed with Proteome Discoverer version 2.5 (Thermo Fisher Scientific) and searched against the mouse and human (for EGFR peptides) SwissProt database using Mascot version 2.4 (RRID:SCR_014322; MatrixScience). MS/MS spectra were searched with mass tolerance of 10 ppm for precursor ions and 20 mmu for fragment ions. Cysteine carbamidomethylation, TMT-labeled lysine, and TMT-labeled peptide N-termini were set as fixed modifications. Oxidation of methionine and phosphorylation of serine, threonine, and tyrosine were searched as dynamic modifications. TMT reporter quantification was extracted and isotope corrected in Proteome Discoverer. Peptide spectrum matches were filtered according to the following parameters: rank = 1, mascot ion score>15, isolation interference<40%, average TMT signal>1,000. Peptides with missing values across any channel were filtered out.

## Phenotypic measurements

### *Proliferation assay*
EGFR mutant and WT-expressing cells were seeded in 96-well tissue culture plastic plates (VWR) at 10,000 cells/well in culture media and allowed to adhere for 24 h. Media were replaced by media containing 1% FBS and 2 nM EGF or 1% FBS and no EGF. Cells were analyzed by a live-cell imaging platform (Incucyte) every 3 h for 48 h. Proliferation rates were calculated using confluence measurements and correlation values between phenotypic measurements were calculated using delta rates with/without EGF stimulation.

### Migration wound scratch assay

EGFR mutant and WT-expressing cells were seeded at $1 \times 10^5$ cells/well in 96-well tissue culture plastic plates (Essen Bioscience) at 100,000 cells/well in culture media and allowed to adhere for 24 h and reach ~90% confluency. The Wound Maker (Essen Bioscience) was used to create a scratch across the well surface. Wells were washed with PBS to remove floating cells and media were added containing 1% FBS and either 2 nM EGF or no EGF. Wound closure was monitored by a live-cell imaging platform (Incucyte with scratch wound cell migration software module) every 3 h for 48 h. Migration rates were calculated using wound width and wound confluency measurements.

### Internalization/receptor trafficking measurements

Biotinylated EGF complexed with pH-sensitive fluorophore pHrodo (Thermo Fisher Scientific) was used to measure receptor internalization for each line. This fluorophore is weakly fluorescent outside the cells at neutral pH but becomes brightly fluorescent in acidic endosomes after EGFR internalization. Cells were plated in six-well plates and cultured to 70–80% confluence. The plates were put on ice for 10 min to inhibit any ongoing endocytosis. The cells were then washed with Live Cell Imaging Solution (LCIS, Thermo Fisher Scientific) supplemented with 20 mM glucose. The cells were then incubated in 250 $\mu$L LCIS/glucose with 0.5 $\mu$g/ml EGF-conjugate for 5 min at 37°C. Then, the cells were washed in cold LCIS, and kept on ice while being removed from the plate using Accutase (Thermo Fisher Scientific). The cells were spun down at 200$g$ for 5 min and washed twice in cold LCIS before resuspending in glucose-supplemented LCIS for analysis by flow cytometry.

### Data analysis

Data analyses were performed in MATLAB R2020A, Microsoft Excel 2016 and Graphpad Prism 9. TMT reporter ion intensities from peptide spectrum matches were summed for each unique phosphopeptide. For protein level quantification, TMT reporter intensities were summed for all unique peptides. Peptide or protein quantification was normalized with relative median values obtained from crude lysate analysis to adjust for sample loading in TMT channels. A combination of $t$ test and one-way ANOVA was used to perform statistical analysis between conditions. Statistical significance was assigned for $P < 0.05$. Unsupervised hierarchical clustering was performed on the basis of Pearson correlation distance metric, unless otherwise specified. Protein networks were obtained from STRING (version 11.0) database.

A SOM was used to cluster proteins that exhibited similar signaling dynamics after EGF treatment. Clustering analysis was performed using the self-organizing map toolbox MATLAB package (http://www.cis.hut.fi/projects/somtoolbox). A 5-by-5 neural network was initiated with a hexagonal lattice structure. The input was the log-2 fold change in phosphorylation after EGF treatment relative to an unstimulated condition for each cell line. The network was randomly initiated and used Euclidean distance as the metric for classifying proteins to specific neurons. The SOM algorithm was repeated 1,000 times, and a co-clustering map was generated indicating the frequency with which any two proteins clustered in the same neuron. This co-clustering map was then subjected to hierarchical clustering using Euclidean distance as the metric for clustering proteins. Biorender was used for schematics. PLSR model was evaluated for goodness of fit ($R^2$) and goodness of prediction ($Q^2$) using leave-one-out-cross validation.

## Data Availability

Proteomic data are available via the PRIDE repository: Project accession: PXD032403.

## Supplementary Information

## Acknowledgements

This research was supported by funding from MIT Center for Precision Cancer Medicine, NIH grants U54 CA210180, U01 CA238720, R01 GM139998, and P30 CA14051.

### Author Contributions

JS Gerritsen: conceptualization, formal analysis, validation, investigation, and writing—original draft, review, and editing.
JS Faraguna: investigation and writing—original draft.
R Bonavia: resources and investigation.
FB Furnari: conceptualization, resources, supervision, and writing—original draft.
FM White: conceptualization, formal analysis, supervision, funding acquisition, project administration, and writing—original draft, review, and editing.

### Conflict of Interest Statement

The authors declare that they have no conflict of interest.

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
