## [Reviewer comments · Life Science Alliance]

Predictive data-driven modeling of C-terminal tyrosine function in the EGFR signaling network

Jacqueline Gerritsen, Joseph Faraguna, Rudy Bonavia, Frank Furnari, and Forest White
DOI: <https://doi.org/10.26508/lsa.202201466>

Corresponding author(s): Forest White, Massachusetts Institute of Technology

Review Timeline:

Submission Date:	2022-03-28
Editorial Decision:	2022-04-28
Revision Received:	2023-02-10
Editorial Decision:	2023-03-14
Revision Received:	2023-04-23
Editorial Decision:	2023-04-24
Revision Received:	2023-04-30
Accepted:	2023-05-01

Transaction Report:

April 28, 2022

Re: Life Science Alliance manuscript #LSA-2022-01466-T

Prof. Forest M White
Massachusetts Institute of Technology
Biological Engineering
77 Massachusetts Avenue, Building 76, Room 353
Cambridge, MA 2139

Dear Dr. White,

Thank you for submitting your manuscript entitled "Predictive data-driven modeling of C-terminal tyrosine function in the EGFR signaling network" to Life Science Alliance. The manuscript was assessed by expert reviewers, whose comments are appended to this letter. We invite you to submit a revised manuscript addressing the Reviewer comments.

Thank you for this interesting contribution to Life Science Alliance. We are looking forward to receiving your revised manuscript.

Sincerely,

B. MANUSCRIPT ORGANIZATION AND FORMATTING:

Reviewer #1 (Comments to the Authors (Required)):

This manuscript examines the phosphorylation of intracellular substrates by EGFR in which individual tyrosine phosphorylation sites have been removed individually or in groups by genetic engineering and expression in receptor-negative mouse fibroblasts. Their basic conclusion is that there is little if any difference in the activity of these receptors, a result that has previously been reported by Linda Pike's group. What is new in this manuscript is the phosphoproteome profile of the cells using multiplexed isobaric tags and well as several biological assays that are distinct from those used by Pike's group. They conclude that although the pattern of protein phosphorylation is very similar between mutant cell lines, the cells differ in their biological responses in several non-intuitive ways. Strengths of the paper include the abundance of high-quality phosphoproteomics data and the examination of biological responses in parallel with the analytical experiments. Weaknesses include the variability in receptor levels between lines, making it difficult to interpret their results. Although their conclusion that Y992 is apparently unique in being associated with the PLC γ pathway, it is difficult to have much confidence in the postulated role of other phosphorylation sites in specific biological activities.

A fundamental problem with these experiments is that the cells are individually derived cell lines that are likely to differ in ways beyond the single sites in the EGFR that were mutated. Mouse fibroblasts are notoriously genetically unstable and different derived lines can show a range of biological differences independent of the expressed transgene. From the data presented in Figure 1B and Supplemental Figure 1, the EGFR expression levels also appear to vary substantially between the different cell lines (e.g., 1045F vs 845F). Expression levels can have a strong impact on several of the biological responses being examined here, such as internalization and growth. For example, it is known that receptor endocytosis is inhibited by high levels of EGFR, which could explain their observation that lines expressing 1045F receptors show low internalization rates. EGF can also inhibit the growth of cells expressing high levels of receptors. This issue is compounded by comparing short term responses (phosphorylation) with long term biological responses that could be significantly impacted by subsequent transcriptional and feedback processes. To demonstrate that the biological differences they observed between lines was indeed due to the receptor isoform rather than clonal variations in the cell lines, a second round of transductions and cell line isolations would need to be done at a minimum.

It is unclear how they define the "EGFR network". Were only tyrosine phosphorylated proteins included? Could some be a product of signaling crosstalk rather than being a direct substrate of the EGFR? For example, it has been shown that >80% of the increase in protein phosphorylation resulting from short treatment times with EGF are downstream of MAPK (DOI 10.1074/mcp.M900285-MCP200). Some of these downstream proteins are tyrosine phosphorylated, the most obvious example being MAPK itself. Thus, if the mutant receptors activate the MAPK pathway, which appears to be the case, then the great majority of the protein phosphorylation pattern should be the same. Since several of their biological assays, such as cell proliferation and migration, are highly dependent on MAPK activation, it is difficult to envision how their results are due to differences in pathway activation rather than biological variations in their cell lines.

I do think their general results are important because they support the idea that there is likely little specific information derived from specific phosphorylation sites. Instead, it seems like the net or density of phosphorylation or the region of the protein that is important. It has been reported that there is evolutionary pressure to balance receptor specificity with adaptor binding (see <https://doi.org/10.1073/pnas.1803598115>). This suggests that spatial relationships might be more important than flanking receptor sequences. I am not sure what "dynamic cellular rewiring" means. Instead of assuming that the system "compensated" for the loss of specific phosphorylation sites, it seems that a simpler explanation is that individual sites have little functional specificity.

Another weakness of the manuscript is a tendency to "cherry-pick" the literature to support their specific hypotheses. For example, the correlation between Gab1 phosphorylation and MAPK activation and endocytosis does not establish any causality between endocytosis and Gab1 phosphorylation. In fact, the rapid kinetics of Gab1 phosphorylation previously reported by this group suggests that Gab1 phosphorylation occurs at the cell surface. PLC γ -1 activation has been shown to occur at both the cell surface and within endosomes (see <https://doi.org/10.1074/jbc.274.13.8958>) and so the negative correlation between internalization and PLC γ -1 phosphorylation is unlikely to be mechanistically connected. The simplest explanation is that secondary factors, such as receptor expression levels or natural variations between cell lines are involved. Considering the lack of experimental testing of any of their correlations, it is advised to downplay them.

Other issues:

1. The pHrodo EGF 488 conjugate measures total internalized EGF, whereas internalization is normally measured as a fraction of surface-associated EGF.

2. Different steps in receptor trafficking are differently impacted by aspects of receptor expression and activity, so a more in-depth analysis is necessary to establish reasonable correlations or causal relationships.
3. The alterations in phosphorylation sites could change the net kinase activity of the receptor. This should be checked
4. The statement "Moderate to high levels of EGFR expression can lead to receptor activation in the absence of ligand." Is not true. A431 cells that expresses millions of EGFR show no receptor activation in the absence of autocrine TGF- α .
5. The relative expression level of the EGFR shown in Figure 1B is difficult to interpret. Is the scale linear? Log? Don't they have proteomics data on the receptor that could be used to calibrate the scale?
6. It was very difficult to read many of the figures because of the very small size of their panels.

Reviewer #2 (Comments to the Authors (Required)):

In this study by Gerritsen and colleagues, the authors generate stable NR6 murine fibroblast cell lines that ectopically express wild-type and mutant EGFR variants and use them to study the functional effects on cellular phosphotyrosine signaling upon EGF stimulation. Growth factor receptor signaling remains a tangled skein of interrelated signaling pathways that drive many basic biological processes and underlie a wide range of human diseases, which provides significance and longitudinal impact for their ongoing study.

This is a well-conceived, well designed and nicely executed study that provides significant insights into downstream signaling in a naïve cellular system. The paper is pretty well written, the figures are excellent and the conclusions are well contained and do not overreach. I have no major reservations about publishing this study.

A few minor things for consideration:

1. Some additional discussion of the possibility of non-native responses to EGF stimulation in an EGFR-naïve cell line would be helpful to better understand the authors' thoughts on how physiologically relevant the NR6 system is to other models of EGF signaling. Does it matter that NR6 cells are from mouse, but the exogenously expressed EGFRs appear to be human?
2. The striking similarities of downstream phosphotyrosine signaling are maybe a bit unexpected given the previously ascribed roles of many of these EGFR phosphotyrosine sites. Perhaps some additional discussion would be helpful to readers on this topic. Could it be the non-native context that regularizes these responses?
3. As a corollary to the previous point, although the authors state that the mutants exhibit "relatively slight changes in surface level expression," it looks like from Fig 1B that there's a ~6-fold difference between the highest and lowest expressing variants. Perhaps it's better to revise the text to simply state the exact range of differences in expression. I'm wondering aloud what the downstream signaling responses would look like in NR6 cells transfected with wtEGFR at 6-fold differences in abundance. Could the authors estimate the absolute amount of wtEGFR in that cell line per cell (or per μ g lysate), such that comparisons could be made to other human, EGFR-competent cell lines? Not critical but I'm wondering if responses become saturated at very high levels of overexpression, and Fig 1B doesn't help in providing this information.
4. Some additional information about the cell lines would be helpful. Are they clonal? Are they polyclonal? If the latter, roughly how many individual clones comprise each cell line? Doesn't have to be exact but a rough estimate would be informative. Super-minor point: the methods should separate "Generation of retrovirus" and "Retroviral transduction." Although 293T cells are transfected to generate retrovirus, NR6 cells are transduced to generate cell lines.
5. Some additional speculation regarding phosphoserine/threonine signaling downstream of these EGFR variants would be welcomed. Is this signaling space perhaps better suited to explain the differences in cell phenotypes for the mutants?
6. Although the main text in the manuscript is correct, the legend in Fig 4A refers to "Principle Component Analysis."

Reviewer #3 (Comments to the Authors (Required)):

In this study Gerritsen et al., report the relationship between phosphorylation of specific EGFR residues to signaling events as well as phenotypic changes. Although there is substantial prior research examining EGFR phosphorylation and signaling, strengths of the current study include state of the art phosphoproteomics methodology and a rigorous data driven approach that adds valuable information to understanding EGFR function. Below are some issues that should be addressed.

1. It would be helpful to have a Western blot of EGFR expression in the various lines. Currently the data are shown in Fig, 1B and Supplemental Fig. 1 as flow cytometry data. The level of EGFR expression itself may impact downstream signaling and phenotypic changes. This could continue to be a confounding factor even with "normalizing" the temporal data. It would also be helpful to see the basal phosphorylation of the EGFR mutants compared to the wild type receptor.
2. The increase in phosphorylation following EGF stimulation for 30 seconds in the parental cells that do not express EGFR is puzzling. Is it possible to exclude a low level of EGFR expression using a murine specific EGFR antibody?
3. A Western blot should be done for the PLCG data shown in Figure 3B.
4. It would be helpful to have some microscopy images of the wound closure experiments.

5. The data suggesting altered phosphorylation of other RTKs (PDGFR, EPHA2) etc and their decrease in the Y992 mutant are quite interesting. Are other RTKs expressed in the NR6 cell line?
6. In results, page 17 "In agreement with our model predictions, phosphorylation of Eps8 Y601 has been reported to block proliferation and to promote cell migration (Cunningham et al, 2013)" The effects of Eps8 in proliferation and cell migration were not reported in Cunningham's paper. Please check.
7. The authors should acknowledge and discuss their work in the context of similar prior studies that have examined the relationship of specific EGFR phosphotyrosine sites, signaling events and phenotypic changes (PMID 1314835, PMID 2022651, PMID 1646987 PMID 2022652)

Reviewer #1 (Comments to the Authors (Required)):

This manuscript examines the phosphorylation of intracellular substrates by EGFR in which individual tyrosine phosphorylation sites have been removed individually or in groups by genetic engineering and expression in receptor-negative mouse fibroblasts. Their basic conclusion is that there is little if any difference in the activity of these receptors, a result that has previously been reported by Linda Pike's group. What is new in this manuscript is the phosphoproteome profile of the cells using multiplexed isobaric tags and well as several biological assays that are distinct from those used by Pike's group. They conclude that although the pattern of protein phosphorylation is very similar between mutant cell lines, the cells differ in their biological responses in several non-intuitive ways. Strengths of the paper include the abundance of high-quality phosphoproteomics data and the examination of biological responses in parallel with the analytical experiments. Weaknesses include the variability in receptor levels between lines, making it difficult to interpret their results.

Response: We appreciate the reviewer's general comments on our manuscript. With regard to variability of receptor expression level across the different cell lines, we attempted to match receptor expression as best as possible for each of the mutant-expressing lines, including selecting high expression cells, maintaining selection pressure throughout cell culture, and performing flow cytometry on each biological replicate that was subsequently used for phosphoproteomics experiments. Despite these efforts, there remains some receptor expression variability across the mutant cell lines, although the variation across most of the lines, with the exception of Y1045F and Y845F, is insignificant when compared to the wild-type expressing line. Since Y845F and Y1045F-expressing lines were the clear outliers in receptor expression, we wanted to determine how these cell lines were impacting the PLSR modeling results. To this end, we reran the PLSR analysis while excluding data from either the Y845F or the Y1045F cell lines, and also re-ran this model excluding data from both Y845F and Y1045F cell lines. The model performed similarly in the presence or absence of these lines, with a slight decrease in Q^2 predictive value for each of the new models relative to the original, as may be expected since we are removing ~10-20% of the data used to create the model. These results suggest that changes in receptor expression among the different lines were not adversely impacting our computational modeling results. These new modeling results have been added to the supplemental data (Supplemental Figure 7b) and the following comment was added to the results section main text:

“To assess the impact of Y845F and Y1045F cell lines on the model, we removed the data from these lines, individually or in combination, and generated a PLSR model from the reduced data set. These new models featured similar fit to data R^2 and predictive Q^2 values, suggesting that Y845F and Y1045F were not adversely affecting model accuracy (Supplemental Fig 7b).”

Although their conclusion that Y992 is apparently unique in being associated with the PLCg pathway, it is difficult to have much confidence in the postulated role of other phosphorylation sites in specific biological activities.

Response: We respectfully disagree with this statement. As noted in the manuscript, many of the predicted associations between phosphorylation site and phenotype are already well established in the literature, thus providing support for the postulated roles of these sites. Indeed, the large number of sites whose predicted function is in

agreement with a priori knowledge suggests that novel predicted functions highlighted by the model are likely to be correct.

A fundamental problem with these experiments is that the cells are individually derived cell lines that are likely to differ in ways beyond the single sites in the EGFR that were mutated. Mouse fibroblasts are notoriously genetically unstable and different derived lines can show a range of biological differences independent of the expressed transgene.

Response: We respectfully disagree with this statement as well. In this study populations of cells were virally infected with each construct, yielding cell lines that stably express each isoform of EGFR. Since viral infection is performed at the population level followed by selection for infected cells, the likelihood that different genetic clones would emerge to dominate individual populations is decreased as compared to generation of clonal cell lines. While we cannot rule out random mutations emerging in some cells, it is highly unlikely that different random mutations emerged in each cell population.

From the data presented in Figure 1B and Supplemental Figure 1, the EGFR expression levels also appear to vary substantially between the different cell lines (e.g., 1045F vs 845F). Expression levels can have a strong impact on several of the biological responses being examined here, such as internalization and growth. For example, it is known that receptor endocytosis is inhibited by high levels of EGFR, which could explain their observation that lines expressing 1045F receptors show low internalization rates.

Respond: We agree with the reviewer's comment, and have noted this biological explanation for the high EGFR expression levels in the 1045F line in the text:

“phosphorylation of the Y1045 site has been associated with binding of the Cbl E3 ubiquitin ligase and subsequent ubiquitination of EGFR. Blocking phosphorylation at this site through Y-to-F mutation should therefore result in altered receptor trafficking, including increased recycling and decreased degradation, providing a potential explanation for the significantly greater cell surface expression relative to wtEGFR”.

Moreover, as noted above, we attempted to match receptor expression levels as best as possible across the different lines and across the biological replicates for each line. In the interest of full transparency, we have included the flow data quantifying receptor expression for all of these conditions (Supplemental Figure 1).

EGF can also inhibit the growth of cells expressing high levels of receptors. This issue is compounded by comparing short term responses (phosphorylation) with long term biological responses that could be significantly impacted by subsequent transcriptional and feedback processes.

Response: We agree that EGF can inhibit growth of cells expressing high levels of receptors; for this reason we quantified phenotypic response for each line under basal and stimulated conditions. While feedback processes are undoubtedly present in these cells, our data in this study suggest that early signaling data is highly predictive of long-term biological responses in these NR6-derived lines. This finding is highly consistent with multiple of our previous

publications connecting immediate-early pTyr signaling to longer-term phenotypic cell response data.

To demonstrate that the biological differences they observed between lines was indeed due to the receptor isoform rather than clonal variations in the cell lines, a second round of transductions and cell line isolations would need to be done at a minimum.

Response: For several of the cell lines generated, transductions were repeated in NR6 cells which yielded similar levels of EGFR expression, indicating that the observed differences were indeed due to the biological effects of receptor mutation.

It is unclear how they define the "EGFR network". Were only tyrosine phosphorylated proteins included? Could some be a product of signaling crosstalk rather than being a direct substrate of the EGFR? For example, it has been shown that >80% of the increase in protein phosphorylation resulting from short treatment times with EGF are downstream of MAPK (DOI 10.1074/mcp.M900285-MCP200). Some of these downstream proteins are tyrosine phosphorylated, the most obvious example being MAPK itself. Thus, if the mutant receptors activate the MAPK pathway, which appears to be the case, then the great majority of the protein phosphorylation pattern should be the same. Since several of their biological assays, such as cell proliferation and migration, are highly dependent on MAPK activation, it is difficult to envision how their results are due to differences in pathway activation rather than biological variations in their cell lines.

Response: We apologize for not clarifying this term. In this work we have specifically focused on tyrosine phosphorylation sites, most of which are altered by EGFR stimulation. We chose to focus on tyrosine phosphorylation sites for this study as it is well established that activation of the EGFR tyrosine kinase leads to altered tyrosine phosphorylation on hundreds of proteins regulating a diverse array of biological activities including proliferation, migration and invasion, metabolic activity, cell morphological changes, focal adhesions, cell-cell contact, among others. While ERK MAP kinase phosphorylation and activity are clearly altered by EGFR activation, this is but one pathway in the tyrosine phosphorylation signaling network. Differential phosphorylation of some of the network components contribute to differences in biological phenotype (migration, proliferation, receptor trafficking) that may depend on, but are not exclusively regulated by ERK MAPK activity.

To clarify these points, we added the following text to the results section:

"Tyrosine phosphorylation sites were quantified in this study as most of the immediate-early signaling network downstream of EGFR activation (e.g., the EGFR network) is regulated by tyrosine phosphorylation, and we have previously shown that this pTyr-mediated network comprises hundreds of sites and is associated with downstream cell phenotypic responses."

With regard to the reference cited above (DOI 10.1074/mcp.M900285-MCP200), that study was performed using 'global phosphoproteomics' which is dominated by pS/pT data, as pSer comprises ~90% of the phosphoproteome and pThr comprises ~9% of the phosphoproteome. As has been shown extensively over the past 15 years, this global phosphoproteomics approach is unable to detect most of the relatively low abundance pTyr phosphorylation sites that form the immediate-early part of the EGFR

signaling network. Thus, the cited manuscript predominantly detected phosphorylation sites downstream of the ERK MAPKs. Give that this previous manuscript did not detect or quantify the early signaling dynamics associated with tyrosine phosphorylation, it is hard to compare their results to ours.

I do think their general results are important because they support the idea that there is likely little specific information derived from specific phosphorylation sites. Instead, it seems like the net or density of phosphorylation or the region of the protein that is important. It has been reported that there is evolutionary pressure to balance receptor specificity with adaptor binding (see <https://doi.org/10.1073/pnas.1803598115>). This suggests that spatial relationships might be more important than flanking receptor sequences.

Response: This is an interesting concept. Recent manuscripts are beginning to quantify the EGFR interactome through APEX and BioID-based approaches. In future studies it will be intriguing to couple some of those methods with our isogenic cell lines containing these different EGFR Y>F mutations to determine whether loss of a given phosphorylation site(s) alters recruitment to the receptor.

I am not sure what "dynamic cellular rewiring" means. Instead of assuming that the system "compensated" for the loss of specific phosphorylation sites, it seems that a simpler explanation is that individual sites have little functional specificity.

Response: We appreciate the reviewer's perspective on this point and apologize for the vague language of "dynamic cellular rewiring". We have removed this term from the text and have re-written this section:

"The strong similarity in network response to EGF stimulation regardless of loss of tyrosine function may be associated with the ability of most of the SH2 and PTB domains on adapter proteins to bind to multiple pTyr sites in the C-terminal tail of the receptor. In the context of loss of one or more sites, these domains may bind to a less favorable site and thus preserve protein-protein interactions and the cellular signaling response. Notably, not all adapter protein domains bind to multiple sites; Cbl appears to have strong preference for Y1045 and PCLg appears to have strong preference for Y992, as indicated by our results and by multiple previous studies."

Another weakness of the manuscript is a tendency to "cherry-pick" the literature to support their specific hypotheses. For example, the correlation between Gab1 phosphorylation and MAPK activation and endocytosis does not establish any causality between endocytosis and Gab1 phosphorylation. In fact, the rapid kinetics of Gab1 phosphorylation previously reported by this group suggests that Gab1 phosphorylation occurs at the cell surface. PLCg-1 activation has been shown to occur at both the cell surface and within endosomes (see <https://doi.org/10.1074/jbc.274.13.8958>) and so the negative correlation between internalization and PLCg-1 phosphorylation is unlikely to be mechanistically connected. The simplest explanation is that secondary factors, such as receptor expression levels or natural variations between cell lines are involved. Considering the lack of experimental testing of any of their correlations, it is advised to downplay them.

Response: We respectfully disagree with this comment. We firmly believe that previously published manuscripts support correlations emerging from the PLSR computational model and thereby strengthen the manuscript rather than weakening it. It is also worth noting that we are careful to not claim causality, as PLSR does not show causality, but associations and correlations. With regard to ‘cherry-picking’, given the large size of the dataset and the large number of associations emerging from the model, we are limited in the number of associations that can be discussed and chose to select some of these points that were well-supported in the literature. Finally, the reviewer is apparently mistaken regarding the conclusions of the cited manuscript. In fact, the abstract of the above cited manuscript clearly states “we found that only cell surface receptors effectively participate in PLC function”, suggesting that the negative correlation between receptor internalization and PLCg phosphorylation is indeed likely to be mechanistically connected.

Other issues:

1. The pHrodo EGF 488 conjugate measures total internalized EGF, whereas internalization is normally measured as a fraction of surface-associated EGF.

Response: We agree with this point. We attempted an alternative strategy using fluorophore-conjugated EGF along with washing steps to get rid of unbound EGF as a measure of surface-associated EGF. However, during our optimization steps we found that the washing conditions washed off some of the cells and hence were not able to get reliable quantification using that method. We could potentially normalize the pHrodo EGF 488 signal for by the ratio of receptor expression as measured by fluorophore conjugated EGF (e.g., the data in Figure 2B), however, as these were measured using different approaches, we were unsure as to whether this would be appropriate for accurate quantification of internalization.

2. Different steps in receptor trafficking are differently impacted by aspects of receptor expression and activity, so a more in-depth analysis is necessary to establish reasonable correlations or causal relationships.

Response: We agree that receptor expression and activity can impact different stages of receptor trafficking. Here we are simply performing a PLSR-based correlation analysis between total internalized receptor and tyrosine phosphorylation site amounts for each cell line, and therefore disagree that a more in-depth analysis is required to establish correlations. At the same time, we agree that causal relationships would clearly require additional mechanistic studies.

3. The alterations in phosphorylation sites could change the net kinase activity of the receptor. This should be checked

Response: We are unaware of an assay to measure EGFR kinase activity in the cell. Alternate strategies would require lysing the cell and performing a kinase activity assay in whole cell lysate or lysing the cell, immunoprecipitating EGFR, and performing a kinase activity assay to yield more specific results. Unfortunately, both of these methods are likely to yield inaccurate results, as EGFR kinase activity can easily be altered by cell lysis disrupting the cell membrane and altering protein-protein interactions.

4. The statement "Moderate to high levels of EGFR expression can lead to receptor activation in the absence of ligand." Is not true. A431 cells that expresses millions of EGFR show no receptor activation in the absence of autocrine TGF- α .

Response: We respectfully disagree with the reviewer's comment, but appreciate that there is a diversity of views on this topic. We have removed this sentence.

5. The relative expression level of the EGFR shown in Figure 1B is difficult to interpret. Is the scale linear? Log? Don't they have proteomics data on the receptor that could be used to calibrate the scale?

Response: We apologize for the difficulty in interpreting this data. The relative expression level of EGFR in Figure 1B is linear, derived from flow plots. We queried our protein expression data, but unfortunately, peptides from EGFR were not detected in these analyses.

6. It was very difficult to read many of the figures because of the very small size of their panels.

Response: We apologize for the difficulty in reading many of the figures, We hope that this will be fixed with the higher quality images of the published version of the manuscript.

Reviewer #2 (Comments to the Authors (Required)):

In this study by Gerritsen and colleagues, the authors generate stable NR6 murine fibroblast cell lines that ectopically express wild-type and mutant EGFR variants and use them to study the functional effects on cellular phosphotyrosine signaling upon EGF stimulation. Growth factor receptor signaling remains a tangled skein of interrelated signaling pathways that drive many basic biological processes and underlie a wide range of human diseases, which provides significance and longitudinal impact for their ongoing study.

This is a well-conceived, well designed and nicely executed study that provides significant insights into downstream signaling in a naïve cellular system. The paper is pretty well written, the figures are excellent and the conclusions are well contained and do not overreach. I have no major reservations about publishing this study.

Response: We thank the reviewer for their kind comments.

1. Some additional discussion of the possibility of non-native responses to EGF stimulation in an EGFR-naïve cell line would be helpful to better understand the authors' thoughts on how physiologically relevant the NR6 system is to other models of EGF signaling. Does it matter that NR6 cells are from mouse, but the exogenously expressed EGFRs appear to be human?

Response: We appreciate this comment, as we evaluated many different cell systems for this study. We attempted to generate point-mutations in human cell lines (MCF10A, A549, and HeLa) through a variety of experimental approaches (CRISPR-based mutagenesis, knock-out of endogenous receptors followed by expression of point mutant receptors) but ran into intractable experimental issues with each system. After perusing the literature and talking to several experts in the field, we selected NR6 cells as a reasonable compromise, given that they have no endogenous

expression of HER-family receptors yet previous studies had shown that they were responsive to exogenously expressed human EGFR (PMID 1933876, 11857451, 9162042, 17363548). With regard to signaling, mouse adapter proteins appear to recognize human EGFR as there are minimal differences in the TKD and C-terminal sequence of EGFR between human and mouse. Moreover, there are virtually no differences in the amino acid sequences directly surrounding the tyrosine sites on the intracellular domain. These points have been added to the manuscript:

“Although NR6 cells do not naturally express EGFR or other ErbB receptors, they are signaling competent, and NR6 cells transfected with human EGFR have been shown to respond to EGF stimulation by increasing proliferation and migration, suggesting intact downstream signaling networks and compatibility of murine cells to expression of human EGFR in agreement with multiple previous studies (Pruss & Herschman, 1977; Jamison et al, 2013; Rosenfeld et al, 1991; Glading et al, 2000). Moreover, mouse adapter proteins appear to recognize human EGFR as there are minimal differences in the TKD and C-terminal sequence of EGFR between human and mouse. Additionally, there are virtually no differences in the amino acid sequences directly surrounding the tyrosine sites on the intracellular domain.”

2. The striking similarities of downstream phosphotyrosine signaling are maybe a bit unexpected given the previously ascribed roles of many of these EGFR phosphotyrosine sites. Perhaps some additional discussion would be helpful to readers on this topic. Could it be the non-native context that regularizes these responses?

Response: We were also initially surprised by the striking similarity among the signaling network response to EGF stimulation across these isogenic lines. Given the sequence similarity (noted above) between human and murine EGFR and human and murine adapter proteins, we believe it is unlikely that the non-native context of the receptor is responsible, although we cannot rule out this point entirely. We added the following sentence to the discussion to mention this possibility:

“It is also possible that the non-native environment of NR6 cells may regularize the signaling network response across different mutant isoforms.”

Instead, it seems more likely that the SH2 and PTB domains of the adapter proteins can bind to multiple sites on the receptor, providing redundancy/robustness/resilience in the signaling network. Loss of one or more tyrosine phosphorylation sites might lead to adapter proteins binding to less favorable sites, but binding nonetheless, therefore yielding highly similar signaling response, even in the context of multiple Y>F mutations. We have added the following statements to the discussion section to capture these points:

“The strong similarity in network response to EGF stimulation regardless of loss of tyrosine function may be associated with the ability of most of the SH2 and PTB domains on adapter proteins to bind to multiple pTyr sites in the C-terminal tail of the receptor. In the context of loss of one or more sites, these domains may bind to a less favorable site and thus preserve protein-protein interactions and the cellular signaling response. Notably, not all adapter protein domains bind to multiple sites; Cbl appears to have strong preference for Y1045 and PCLg appears to have strong preference for Y992, as indicated by our results and by multiple previous studies.”

3. As a corollary to the previous point, although the authors state that the mutants exhibit "relatively slight changes in surface level expression," it looks like from Fig 1B

that there's a ~6-fold difference between the highest and lowest expressing variants. Perhaps it's better to revise the text to simply state the exact range of differences in expression. I'm wondering aloud what the downstream signaling responses would look like in NR6 cells transfected with wtEGFR at 6-fold differences in abundance. Could the authors estimate the absolute amount of wtEGFR in that cell line per cell (or per ug lysate), such that comparisons could be made to other human, EGFR-competent cell lines? Not critical but I'm wondering if responses become saturated at very high levels of overexpression, and Fig 1B doesn't help in providing this information.

Response: We appreciate this suggestion and have modified the text accordingly. New modified text:

“Most of the mutant isoforms were expressed at a similar level to wtEGFR, with the exception of Y845F and Y1045F, expressed at 0.5x and 3x of the levels of wtEGFR (Fig 1B).”

To estimate receptor expression levels, we compared expression levels to other lines with known copy number (A549, A431, MCF10A) and found levels in our transfected NR6 cells to be similar to A549 cells in the vicinity of ~150,000 copies per cell (Jaramillo Maria L, Leon Zully, Grothe Suzanne, Paul-Roc Beatrice, Abulrob Abedelnasser, 2006). From our reading of the literature, it appears HeLa cells have approximately 50K copies/cell, A431 cells have ~1-2M copies, and MCF10A have ~10K copies/cell. This comparison has been added to the manuscript as supplemental Figure 1B, and the following text has been added to describe these results:

“By comparison with other cell lines with known levels of EGFR expression, we determined that wtEGFR was expressed at ~150,000 copies per cell in our transfected NR6 lines (Supplemental Figure 1B).”

4. Some additional information about the cell lines would be helpful. Are they clonal? Are they polyclonal? If the latter, roughly how many individual clones comprise each cell line? Doesn't have to be exact but a rough estimate would be informative. Super-minor point: the methods should separate "Generation of retrovirus" and "Retroviral transduction." Although 293T cells are transfected to generate retrovirus, NR6 cells are transduced to generate cell lines.

Response: The cell lines were generated by retroviral transduction of a large population of cells (~400,000 cells), and thus comprise a large population of individual

clones; about 10,000 cells were sorted per line. We attempted to generate clonal lines, but individual clones would not survive – apparently these cells may rely on growth factors from neighboring cells? Methods were separated as suggested.

5. Some additional speculation regarding phosphoserine/threonine signaling downstream of these EGFR variants would be welcomed. Is this signaling space perhaps better suited to explain the differences in cell phenotypes for the mutants?

Response: We chose to focus on pTyr signaling, as previous manuscripts from our group and others (PMID: 17081983, 17206861, 17016520, 23438512, 26929352) have indicated that EGFR activation has a very strong effect on the pTyr signaling network (>70% of pTyr sites can be altered by EGFR activation), while only ~<10% of pSer/pThr sites are altered following EGFR activation. Moreover, as noted by Reviewer 1, above, much of the pS/pT changes are mediated by MAPK activity and thus are less reflective of the cell morphology, cell-cell contact, focal adhesions, etc. that regulate cell migration. While we agree that pS/pT data may be additionally informative, intriguingly, our PLSR models based on phosphotyrosine data were found to be very predictive of phenotype with Q2 of over 0.9, suggesting that pTyr may be highly relevant for these downstream cell responses. We added the following comment to the results section to clarify these points:

“Additional insight may be gained by quantifying and modeling serine and threonine phosphorylation (pS/pT) in these systems, but much of this signaling lies downstream of the pTyr-mediated signaling network and the vast majority of pS/pT sites are unaltered by EGF stimulation.”

6. Although the main text in the manuscript is correct, the legend in Fig 4A refers to "Principle Component Analysis."

Response: Thanks! – we have fixed this error.

Reviewer #3 (Comments to the Authors (Required)):

In this study Gerritsen et al., report the relationship between phosphorylation of specific EGFR residues to signaling events as well as phenotypic changes. Although there is substantial prior research examining EGFR phosphorylation and signaling, strengths of the current study include state of the art phosphoproteomics methodology and a rigorous data driven approach that adds valuable information to understanding EGFR function.

Response: We appreciate these kind remarks regarding the potential impact of this study.

1. It would be helpful to have a Western blot of EGFR expression in the various lines. Currently the data are shown in Fig, 1B and Supplemental Fig. 1 as flow cytometry data. The level of EGFR expression itself may impact downstream signaling and phenotypic changes. This could continue to be a confounding factor even with "normalizing" the temporal data. It would also be helpful to see the basal phosphorylation of the EGFR mutants compared to the wild type receptor.

Response: We have generated western blots of EGFR expression for each of the various lines in the absence and presence of ligand stimulation. It is worth noting that our flow cytometry data measures surface EGFR capable of binding EGF ligand, while the western blot data provides a measure of total EGFR. These data are now included in Supplemental figure 4 for basal (unstimulated) phosphorylation conditions of mutants compared to wild type.

2. The increase in phosphorylation following EGF stimulation for 30 seconds in the parental cells that do not express EGFR is puzzling. Is it possible to exclude a low level of EGFR expression using a murine specific EGFR antibody?

Response: We agree that this result is puzzling. We speculate that the addition of EGF in vehicle followed by a gentle swirl of the media in the plate may lead to cell signaling activation due to mechanical / shear stress. This signaling is likely overwhelmed by EGFR activation and resultant signaling in cells expressing EGFR.

Unfortunately, a murine specific anti-EGFR antibody is not available, as all antibodies recognize mouse as well as human EGFR (although there are some that “prefer” mouse). However, by multiple measurements, including our labeled-EGF flow data, and western blots using anti-EGFR antibodies, the non-transfected cells show a lack of EGFR signal. Lastly, it is also worth noting that our phosphoproteomics data does not show any EGFR expression or EGFR phosphorylation in the untransfected lines.

3. A Western blot should be done for the PLCG data shown in Figure 3B.

Response: We have attempted this experiment multiple times. While the total PLCg blot has worked well, unfortunately the phospho-PLCg blot has not worked well, likely due to the polyclonal nature of the commercially available anti-pY1253 PLCg antibodies. It is worth noting that many ‘phospho-specific’ antibodies recognizing specific pTyr sites are notoriously poor – we have approximately a 20% success rate in using these commercial antibodies over the past 15 years.

4. It would be helpful to have some microscopy images of the wound closure experiments.

Response: We thank the reviewer for this helpful suggestion. While we would

have loved to accommodate this request, our Incucyte unfortunately suffered a 'RAID degradation' resulting in loss of the microscopy images for these wound closure experiments.

5. The data suggesting altered phosphorylation of other RTKs (PDGFR, EPHA2) etc and their decrease in the Y992 mutant are quite interesting. Are other RTKs expressed in the NR6 cell line?

Response: We agree that the altered phosphorylation of other RTKs in the Y992F-expressing cell line is intriguing. The NR6 cells express multiple RTKs, including but not limited to PDGFR-alpha, AXL, FGFR1, EphA2, EphA7, EPHB4.

6. In results, page 17 "In agreement with our model predictions, phosphorylation of Eps8 Y601 has been reported to block proliferation and to promote cell migration (Cunningham et al, 2013)" The effects of Eps8 in proliferation and cell migration were not reported in Cunningham's paper. Please check.

Response: We apologize for this oversight and have changed the citation to Cattaneo et al 2012 and Ding et al, 2013. (Cattaneo Maria Grazia, Cappellini Elisa, 2012; Ding Xiaofeng, Zhou Fangliang, Wang Fangmei, Yang Zijian, Zhou Chang, Zhou Jianlin, Zhang Bo, Yang Junmei, Wang Guangwei, Wei Zheng, Hu Xiang, Xiang Shuanglin, Zhang, 2013)

7. The authors should acknowledge and discuss their work in the context of similar prior studies that have examined the relationship of specific EGFR phosphotyrosine sites, signaling events and phenotypic changes (PMID 1314835, PMID 2022651, PMID 1646987 PMID 2022652)

Response: We appreciate this suggestion and have added the following text to the introduction section:

"Other studies have related specific EGFR tyrosine phosphorylation sites to cell phenotypic response, including those that have highlighted the role of Y992 and Y1045 in regulating EGFR trafficking (PMID 1314835, PMID 2022651, PMID 1646987 PMID 2022652)."

We also added the following comments to the discussion section:

"Previous work utilizing cell and molecular biological approaches has provided connections between selected phosphorylation sites and downstream cell phenotypic response to stimulation (PMID 1314835, PMID 2022651, PMID 1646987 PMID 2022652). These studies required a priori knowledge of the phosphorylation sites to be interrogated and had to be performed through mutagenesis of each phosphorylation site. Here, through proteomics and computational modeling, we were able to generate predicted associations for many phosphorylation sites to each phenotype. Follow-on studies to interrogate the mechanistic connections for each of these predicted associations are still required and are likely to yield novel insight into signaling network regulation underlying cell decision processes following receptor stimulation."

March 14, 2023

Re: Life Science Alliance manuscript #LSA-2022-01466-TR

Prof. Forest M White
Massachusetts Institute of Technology
Biological Engineering
77 Massachusetts Avenue, Building 76, Room 353
Cambridge, MA 2139

Dear Dr. White,

Thank you for submitting your revised manuscript entitled "Predictive data-driven modeling of C-terminal tyrosine function in the EGFR signaling network" to Life Science Alliance. The manuscript has been seen by the original reviewers whose comments are appended below. While the reviewers continue to be overall positive about the work in terms of its suitability for Life Science Alliance, some important issues remain. Please note that Reviewer 1 provided a figure to emphasize some of their remaining points. This figure is attached to this email.

Our general policy is that papers are considered through only one revision cycle; however, given that the suggested changes are relatively minor, we are open to one additional short round of revision. Please note that I will expect to make a final decision without additional reviewer input upon re-submission.

Please submit the final revision within one month, along with a letter that includes a point by point response to the remaining reviewer comments.

To upload the revised version of your manuscript, please log in to your account: <https://lsa.msubmit.net/cgi-bin/main.plex>
You will be guided to complete the submission of your revised manuscript and to fill in all necessary information.

B. MANUSCRIPT ORGANIZATION AND FORMATTING:

Sincerely,

Reviewer #1 (Comments to the Authors (Required)):

This revised paper has addressed many of the issues of the original manuscript, primarily by providing more information on how the cell lines were generated. I now know that they are analyzing polyclonal populations, which is a reasonable approach and addresses many of my original concerns. It's good that they repeated some of their transductions and got similar results. The consistency of the different surface expression and the somewhat different pattern from the western blots is strongly suggestive of different trafficking patterns, which makes sense considering that trafficking is highly dependent on receptor phosphorylation state. Because these are polyclonal cell lines, clonal variations are highly unlikely to be responsible for their biological differences. Instead, it is more likely that differences in receptor distribution and trafficking are involved. Unfortunately, as the authors have stated, understanding how trafficking is altered in the mutants and relating this to differences in biological responses would require a far more extensive study that is probably not practical.

I do think that the results are interesting because of the lack of clear correlations between specific phosphorylation sites and downstream protein phosphorylation, which they correctly highlight. I was thus puzzled to see them try to assign a causal role to differential phosphorylation of several specific substrates that show no obvious correlation with any biological responses. I just don't find their interpretation of the role of PLC-g1 and EPS8 in migration and proliferation credible. They show that the different receptor mutants display highly similar patterns of initial phosphorylation, but display distinct trafficking patterns, perhaps tied to either conformational changes or substrate phosphorylation. The physiological responses of cells to EGF that they measured take hours to days to develop during which the receptors can undergo different extents of down-regulation, feedback regulation, and perhaps induce differential gene expression, which can then alter the pattern of feedback. Yet they propose that differences in EPS8 phosphorylation at 1-5 minutes is a major contributor to differences between the mutant receptors. This makes me doubt the validity of their PLSR analysis and suggests that the data is too noisy to make firm conclusions. For example, if you look at the kinetics or extent of EPS8 phosphorylation from their supplemental table, it is essentially identical between the Y992F and DY5 mutants that show the largest difference in migration rates. The only real difference in EPS8 phosphorylation is for the Y845F mutation that shows no significant difference in either proliferation or migration. How can EPS8 be a central regulator if pronounced differences in its phosphorylation has no impact on the cells? I also do not understand why they state "Eps8 was strong (sic) correlated with the 1068 residue" when only the Y845F mutant showed any apparent change in its phosphorylation. Perhaps the data in the table is wrong? Alternately, their analysis could be giving them spurious answers. I think the extensive discussion of the significance of the PLSR analysis is not warranted unless there is at least some supporting data gathered in these cell lines (not "cherry picked" from the literature). At the very least, the language supporting the conclusions should be moderated. After all, how can they claim "we were able to generate robust and highly predictive models for each phenotype" when they don't actual show that their models predict anything. They are showing correlations, not predictions.

There are still a few minor issues that need to be addressed. Figure 2 is still hard to read. Figure 4 is very difficult to understand because the ordering and colors of the mutants are different, making it very difficult to visually compare the cell lines. I understand that the ordering might differ for statistical comparison, but please keep the same colors for each cell line. There are a number of typos that should be corrected. For example, "we determined that wtEGFR was expressed at ~150,000 copies be cell in our transfected NR6 lines" should be "we determined that wtEGFR was expressed at ~150,000 copies per cell in our transfected NR6 lines". "NR6 cells expressing wild-type exhibit signaling network..." should be "NR6 cells expressing wild-type receptors exhibit signaling network...". Many such errors.

Reviewer #3 (Comments to the Authors (Required)):

The authors have satisfactorily addressed most of the issues raised in my previous review and I am happy to recommend publication of this interesting work.

We would like to thank the editor and reviewers for taking the time to consider our revised manuscript and provide constructive feedback. Responses to reviewer comments are provided below.

Reviewer 1:

This revised paper has addressed many of the issues of the original manuscript, primarily by providing more information on how the cell lines were generated. I now know that they are analyzing polyclonal populations, which is a reasonable approach and addresses many of my original concerns. It's good that they repeated some of their transductions and got similar results. The consistency of the different surface expression and the somewhat different pattern from the western blots is strongly suggestive of different trafficking patterns, which makes sense considering that trafficking is highly dependent on receptor phosphorylation state. Because these are polyclonal cell lines, clonal variations are highly unlikely to be responsible for their biological differences. Instead, it is more likely that differences in receptor distribution and trafficking are involved. Unfortunately, as the authors have stated, understanding how trafficking is altered in the mutants and relating this to differences in biological responses would require a far more extensive study that is probably not practical.

I do think that the results are interesting because of the lack of clear correlations between specific phosphorylation sites and downstream protein phosphorylation, which they correctly highlight. I was thus puzzled to see them try to assign a **causal role** to differential phosphorylation of several specific substrates that show no obvious correlation with any biological responses.

Response: We respectfully disagree with this statement. Throughout the manuscript we have attempted to be consistent with our language indicating the PLSR model provides associations or correlations and does not provide causality.

Reviewer 1: I just don't find their interpretation of the role of PLC-g1 and EPS8 in migration and proliferation credible. They show that the different receptor mutants display highly similar patterns of initial phosphorylation, but display distinct trafficking patterns, perhaps tied to either conformational changes or substrate phosphorylation. The physiological responses of cells to EGF that they measured take hours to days to develop during which the receptors can undergo different extents of down-regulation, feedback regulation, and perhaps induce differential gene expression, which can then alter the pattern of feedback. Yet they propose that differences in EPS8 phosphorylation at 1-5 minutes is a major contributor to differences between the mutant receptors.

Response: We respectfully disagree with these statements as well. We and others have previously shown that signaling occurring in the seconds-to-minutes timeframe has a direct impact on the cell decision processes that ultimately lead to cell phenotypic responses in the hours-to-days timeframe. We have added references to previous publications and also altered the following sentence in the discussion section to highlight this point:

“Although cellular responses such as migration and proliferation take hours-to-days to occur, phosphorylation changes in the immediate-early timepoints, as early as 30 seconds or 1 minute, were found to be highly important in predictive models for these longer-term responses, in agreement with previous models of HER-family signaling networks.”

Reviewer 1: This makes me doubt the validity of their PLSR analysis and suggests that the data is too noisy to make firm conclusions.

Response: Again, we respectfully disagree with this statement. Indeed, due to the nature of the PLSR algorithm, noisy data is typically poorly correlated with phenotypic outcome and is therefore excluded from the high scoring VIP nodes.

Reviewer 1: For example, if you look at the kinetics or extent of EPS8 phosphorylation from their supplemental table, it is essentially identical between the Y992F and DY5 mutants that show the largest difference in migration rates. The only real difference in EPS8 phosphorylation is for the Y845F mutation that shows no significant difference in either proliferation or migration. How can EPS8 be a central regulator if pronounced differences in its phosphorylation has no impact on the cells? I also do not understand why they state "Eps8 was strong (sic) correlated with the 1068 residue" when only the Y845F mutant showed any apparent change in its phosphorylation. Perhaps the data in the table is wrong? Alternately, their analysis could be giving them spurious answers.

Response: Unfortunately, the reviewer has overstated this point. The difference in Eps8 phosphorylation is most evident in the early time points, with some of the lines having an increase of over 4-fold ($2 \log_2$) in the first 30 seconds, and other lines having less than a 2-fold change at this same time point. While most lines reach approximately the same level of Eps8 phosphorylation by 5 minutes, the kinetics for several of the lines are significantly different. PLSR models assess correlation between X-matrices and Y-matrices; in our case the X-matrix consisted of phosphorylation levels across all lines and time points while the Y-matrix was comprised of the phenotypic response for each of the cell lines. While our analysis suggests that Eps8 is correlated with proliferation and migration, we do not state that this is the only factor in these cells that is driving this cell phenotype decision. Picking out a single line and noting that Eps8 does not appear to be consistent with phenotype for that line is poor practice and suggests a fundamental misunderstanding of the nature of the PLSR algorithm. We strongly disagree with the statement that the analysis could be providing spurious answers, especially since our model predictions agree with published functions for most, if not all, of the sites that were found to be correlated with proliferation and/or migration.

Reviewer 1: I think the extensive discussion of the significance of the PLSR analysis is not warranted unless there is at least some supporting data gathered in these cell lines (not "cherry picked" from the literature). At the very least, the language supporting the conclusions should be moderated. After all, how can they claim "we were able to generate robust and highly predictive models for each phenotype" when they don't actual show that their models predict anything. They are showing correlations, not

predictions.

Response: PLSR models are, by their nature, quantitative and predictive. Through the model, it is possible to predict the phenotypic change that occurs by removal of a given signaling node or cell perturbation, as has been shown extensively in the literature. With that said, we have moderated the language in the results and discussion section, replacing 'predicted' with 'associated' or 'correlated' where appropriate.

Reviewer 1: There are still a few minor issues that need to be addressed. Figure 2 is still hard to read. Figure 4 is very difficult to understand because the ordering and colors of the mutants are different, making it very difficult to visually compare the cell lines. I understand that the ordering might differ for statistical comparison, but please keep the same colors for each cell line.

There are a number of typos that should be corrected. For example, "we determined that wtEGFR was expressed at ~150,000 copies be cell in our transfected NR6 lines" should be "we determined that wtEGFR was expressed at ~150,000 copies per cell in our transfected NR6 lines". "NR6 cells expressing wild-type exhibit signaling network..." should be "NR6 cells expressing wild-type receptors exhibit signaling network...". Many such errors.

Response: We thank the reviewer for noting these minor issues and have attempted to correct them in the latest version of the manuscript.

April 24, 2023

RE: Life Science Alliance Manuscript #LSA-2022-01466-TRR

Prof. Forest M White
Massachusetts Institute of Technology
Biological Engineering
77 Massachusetts Avenue, Building 76, Room 353
Cambridge, MA 2139

Dear Dr. White,

Thank you for submitting your revised manuscript entitled "Predictive data-driven modeling of C-terminal tyrosine function in the EGFR signaling network". We would be happy to publish your paper in Life Science Alliance pending final revisions necessary to meet our formatting guidelines.

- please add the supplemental figure legends to the main manuscript text
- please add the Twitter handle of your host institute/organization as well as your own or/and one of the authors in our system
- please make dataset PXD032403 publicly accessible at this time, and you can remove the Reviewer access info from your Data Availability statement

Figure Check:

- please add sizes next to the blots in Figure S4B
- please rename Figure S7 as Supplemental Table 1, and update the callouts in the text accordingly

A. FINAL FILES:

B. MANUSCRIPT ORGANIZATION AND FORMATTING:

Thank you for your attention to these final processing requirements. Please revise and format the manuscript and upload materials within 3 days.

Sincerely,

May 1, 2023

RE: Life Science Alliance Manuscript #LSA-2022-01466-TRRR

Prof. Forest M White
Massachusetts Institute of Technology
Biological Engineering
77 Massachusetts Avenue, Building 76, Room 353
Cambridge, MA 2139

Dear Dr. White,

Thank you for submitting your Resource entitled "Predictive data-driven modeling of C-terminal tyrosine function in the EGFR signaling network". It is a pleasure to let you know that your manuscript is now accepted for publication in Life Science Alliance. Congratulations on this interesting work.

DISTRIBUTION OF MATERIALS:

Again, congratulations on a very nice paper. I hope you found the review process to be constructive and are pleased with how the manuscript was handled editorially. We look forward to future exciting submissions from your lab.

Sincerely,
